# Telomere length sensitive regulation of interleukin receptor 1 type 1 (IL1R1) by the shelterin protein TRF2 modulates immune signalling in the tumour microenvironment

Ananda Kishore Mukherjee[1,2‡], Subhajit Dutta[1,2†], Ankita Singh[1†], Shalu Sharma[1,2], Shuvra Shekhar Roy[1,2], Antara Sengupta[1,2], Megha Chatterjee[1], Soujanya Vinayagamurthy[1,2], Sulochana Bagri[1,2], Divya Khanna[1], Meenakshi Verma[1], Dristhi Soni[1,2], Anshul Budharaja[3], Sagar Kailasrao Bhisade[4], Vivek Anand[2,5], Ahmad Perwez[1], Nija George[6], Mohammed Faruq[2,5,7], Ishaan Gupta[3,4], Radhakrishnan Sabarinathan[7], Shantanu Chowdhury[1,2,7,8*]

[1]Integrative and Functional Biology Unit, CSIR-Institute of Genomics and Integrative Biology, New Delhi, India; [2]Academy of Scientific and Innovative Research (AcSIR), Ghaziabad, India; [3]IIT Delhi, Delhi, India; [4]IISER Bhopal, Bhopal, India; [5]Genomics and Molecular Medicine, CSIR-Institute of Genomics and Integrative Biology, New Delhi, India; [6]National Centre for Biological Sciences, Tata Institute of Fundamental Research, Bangalore, India; [7]GNR Knowledge Centre for Genome and Informatics, CSIR-Institute of Genomics and Integrative Biology, New Delhi, India; [8]Trivedi School of Biosciences, Ashoka University, Sonepat, India

**\*For correspondence:**
shantanuc@igib.in

†These authors contributed equally to this work

**Present address:** ‡Kennedy Institute of Rheumatology, NDORMS, University of Oxford, Oxford, United Kingdom

## eLife Assessment

This study presents an **important** finding on the role of telomeres in modulating interleukin-1 signaling and tumor immunity in TNBC. The evidence supporting these findings is **solid**, presented through comprehensive analyses including TNBC clinical samples, tumor-derived organoids, cancer cells, and xenografts. The work will be of broad interest to cell and medical biologists focusing on TNBC.

**Abstract** Telomeres are crucial for cancer progression. Immune signalling in the tumour microenvironment has been shown to be very important in cancer prognosis. However, the mechanisms by which telomeres might affect tumour immune response remain poorly understood. Here, we observed that interleukin-1 signalling is telomere-length dependent in cancer cells. Mechanistically, non-telomeric TRF2 (telomeric repeat binding factor 2) binding at the IL-1-receptor type-1 (IL1R1) promoter was found to be affected by telomere length. Enhanced TRF2 binding at the *IL1R1* promoter in cells with short telomeres directly recruited the histone-acetyl-transferase (HAT) p300, and consequent H3K27 acetylation activated IL1R1. This altered NF-kappa B signalling and affected downstream cytokines like *IL6*, *IL8*, and *TNF*. Further, *IL1R1* expression was telomere-sensitive in triple-negative breast cancer (TNBC) clinical samples. Infiltration of tumour-associated macrophages (TAM) was also sensitive to the length of tumour cell telomeres and highly correlated with *IL1R1* expression. The use of both IL1 Receptor antagonist (IL1RA) and *IL1R1* targeting ligands could abrogate M2 macrophage infiltration in TNBC tumour organoids. In summary, using TNBC cancer tissue

(>90 patients), tumour-derived organoids, cancer cells, and xenograft tumours with either long or short telomeres, we uncovered a heretofore undeciphered function of telomeres in modulating IL1 signalling and tumour immunity.

## Introduction

Telomeres are composed of DNA repeats (TTAGGG) and associated 'shelterin' proteins present at the end of each eukaryotic chromosome. Gradual shortening of telomeres and change in the telomeric structure with cell division leads to cellular senescence/death, which is crucial for tissue homeostasis (*Jacobs, 2013*; *Karlseder et al., 2002*; *Shay and Wright, 2000*). During oncogenic transformation, the telomere shortening mediated cellular senescence is bypassed - resulting in uncontrolled cell proliferation and cancer progression; underlining the impact of telomeres in cancer (*Blackburn, 2000*; *Hanahan and Weinberg, 2000*; *Maciejowski and de Lange, 2017*; *Paeschke et al., 2010*; *Shay, 2016*).

Shortening of telomeres with ageing contributes to compromised immune response (*Goronzy et al., 2006*; *Hodes et al., 2002*; *Pawelec et al., 2014*; *Sansoni et al., 2008*; *Weng, 2012*). However, despite its relevance in cancer prognosis and intervention (*Artandi and DePinho, 2010*; *Aviv et al., 2017*; *Chew et al., 2012*; *Balkwill and Mantovani, 2001*; *Greider, 1998*; *Hanahan and Weinberg, 2011*; *Wang et al., 2017*), how telomeres influence tumour immune response is poorly understood.

Major advances show how immune cells including tumour-infiltrating-lymphocytes (TIL), tumour-associated-macrophages (TAM), natural-killer, and dendritic cells impact cancer progression (*Binnewies et al., 2018*; *Gonzalez et al., 2018*; *Jochems and Schlom, 2011*; *Salgado and Gevaert, 2021*). TAMs – typically M2 macrophage-like – predominantly promote immunosuppressive tumour microenvironment (TME) through secretion of anti-inflammatory cytokines (*Burkholder et al., 2014*; *Cassetta et al., 2016*; *DeNardo and Ruffell, 2019*; *Gonzalez et al., 2018*; *Pittet et al., 2022*; *Jayasingam et al., 2019*), implicating the role of TAMs in aggressive tumour growth (*Hu et al., 2016*; *Pittet et al., 2022*).

Poor prognosis in triple-negative breast cancers (TNBC; negative for estrogen receptor [ER-], progesterone receptor [PR-], and HER2 receptor [HER2-]) has been frequently associated with TAMs (*Wang et al., 2022*; *Yuan et al., 2014*). Multiple reports show M2-like TAMs to be the most abundant monocytic immune cells in TNBC and implicate their role in invasiveness and aggressive tumour growth (*Burkholder et al., 2014*; *Jeong et al., 2019*; *Mantovani and Locati, 2013*; *Medrek et al., 2012*; *Oner et al., 2020*; *Chen et al., 2005*; *Yuan et al., 2014*).

How the telomere length (TL) of cancer cells affects macrophage infiltration has not been studied. Seimiya. H and colleagues noted that telomere elongation led to differential expression of genes associated with innate immunity (*Hirashima et al., 2013*). A published work on colorectal cancer showed the association of TL of cancer cells to immunity-related genes (*Lopez-Doriga et al., 2018*). Another recent article showed that telomere maintenance by Telomerase (the key enzyme responsible for telomere maintenance) and ATL (alternative telomere lengthening) related mechanisms led to alteration of markers of TAM like CD163 in glioblastomas (*Hung et al., 2016*). Taken together, these suggest the possibility of molecular links between TAM infiltration and telomeres.

TRF2 (telomeric repeat binding factor 2) is a canonical member of the telomere capping complex called 'shelterin' with reported extra-telomeric functions (*Martínez and Blasco, 2011*; *Vinayagamurthy et al., 2020*). We have reported earlier that the telomeric protein TRF2 alters gene expression in a TL-dependent fashion (*Mukherjee et al., 2018*). TRF2, although primarily a telomere-binding protein, was found to associate with gene promoters throughout the genome independent of its telomere-specific functions (*Mukherjee et al., 2019a*; *Simonet et al., 2011*; *Yang et al., 2011*). Interestingly, TRF2 was sequestered at the telomeres and away from gene promoters in cells with relatively longer telomeres, and vice-versa in the case of cells with shorter telomeres. This led to TRF2-mediated transcriptional regulation that was dependent on telomere length (a scheme of this model shown in *Figure 1—figure supplement 1*; *Mukherjee et al., 2018*; *Vinayagamurthy et al., 2023*; *Vinayagamurthy et al., 2020*).

We found that non-telomeric TRF2 transcriptionally activated interleukin-1-receptor type-1 (IL1R1) in a telomere-dependent fashion: resulting NF-kappaB (p65)-phosphorylation led to the upregulation of downstream cytokines. Telomere-sensitive IL1 signalling in TNBC was found to be correlated with

enhanced TAM infiltration within the TME in tumours with relatively short telomeres. Together, these implicate telomere-sensitive immune-microenvironment in tumours.

## Results

### Non-telomeric TRF2 binding at the IL1 receptor type 1 (*IL1R1*) promoter is telomere length dependent

#### Cytokine and their receptor-related genes are telomere-sensitive in cancer cells

We performed an RNA-seq experiment using our previously reported fibrosarcoma model of telomere elongation (*Mukherjee et al., 2018*). The transcriptomic profile of HT1080 cells (short telomere cells) was compared with HT1080-LT (long telomere cells). The difference in telomere length between the cells was confirmed using flow cytometry (*Figure 1—figure supplement 2A*). The HT1080-LT cells had constitutively higher expression of *TERT* and *TERC* as previously described (*Figure 1—figure supplement 2B*). The telomere length of the cells was further confirmed using in situ hybridization of telomeric probes (*Figure 1—figure supplement 2C*). Upon analyzing the RNA-seq data, we observed that several cytokine-related genes like *IL1B*, *IL1R1*, *TNF*, *IL15,* and *IL37* had higher expression in HT1080 (short telomere) cells (*Figure 1—figure supplement 2D*). We looked at the enriched pathways (KEGG) in genes that were upregulated (twofold or higher expression) and down-regulated (0.5-fold or lower expression) in HT1080 cells (short telomere) compared to HT1080-LT (long telomere) cells (*Figure 1—figure supplement 2E and F*; *Supplementary file 1*). It was observed that 'cytokine- cytokine receptor interaction' was one of the top 5 most enriched pathways for the upregulated genes (*Figure 1—figure supplement 2E*). While telomere elongation affected multiple pathways, we were most interested in cytokine-related genes that were differentially expressed due to the now well-established focus on the telomere-immunity axis (*Barthel et al., 2017*; *Goronzy et al., 2006*; *Hirashima et al., 2013*; *Weng, 2012*; *Ye et al., 2014*). We revisited our published ChIP-seq data in HT1080 fibrosarcoma cells (*Mukherjee et al., 2019a*) and found multiple cytokine-related gene promoters where TRF2 had peaks within 1000 bp from the transcriptional start site (TSS; *Supplementary file 2*). We validated these sites by ChIP-qPCR in HT1080 fibrosarcoma cells and found that the *IL1R1* promoter had the most prominent TRF2 enrichment among the sites (*Figure 1—figure supplement 2G*). We further performed ChIP-qPCR spanning +200 to –1000 bp of the *IL1R1* TSS to confirm promoter TRF2 binding across multiple cell lines- HT1080 fibrosarcoma, MDAMB231 breast cancer, HEK293T immortalised kidney and MRC5 fibroblast cells (*Figure 1—figure supplement 2H*).

#### Telomere-dependent IL1R1 regulation in ex vivo fibrosarcoma cells

Next, we tested the telomere dependence of TRF2 binding based on TL-sensitive non-telomeric TRF2 binding observed earlier (*Mukherjee et al., 2018*). TRF2-ChIP in ex vivo cultured HT1080-LT/HT1080 cells showed that TRF2 occupancy at the *IL1R1* promoter was significantly reduced in cells with long telomeres relative to short telomere cells (*Figure 1A*). *IL1R1* mRNA expression was significantly lower in HT1080-LT relative to HT1080 cells (*Figure 1B*) conforming to the RNA-seq data described earlier (*Figure 1—figure supplement 2D*). We found that the IL1R1 protein was also lower in HT1080-LT cells (flow cytometry; *Figure 1C*; western blot in *Figure 1—figure supplement 2I*). Consistent with this, expression of IL1R1 at the cell surface was significantly lower in HT1080-LT compared to HT1080 cells (*Figure 1D*).

#### Telomere-dependent IL1R1 regulation in xenograft tumours

We generated xenograft tumours using HT1080 and HT1080-LT cells in NOD-SCID mice (see methods for details). We confirmed that there was higher expression of *TERT* and *TERC* in the HT1080-LT xenograft tumours (*Figure 1—figure supplement 2J*). We found that the xenograft tumours from HT1080-LT cells had higher mean telomere length than those from HT1080 cells and had lower IL1R1 expression at the protein level (*Figure 1E*). It was observed that TRF2 occupancy (*Figure 1F*), as well as *IL1R1* mRNA (*Figure 1G*) expression, was lower in HT1080-LT-derived xenograft tumours.

### TERT-inducible model of telomere elongation in fibrosarcoma cells

Telomere-dependent TRF2 binding at the *IL1R1* promoter was further checked using *hTERT*-inducible HT1080 cells that gradually increased telomeres by ~50% over 12 days following hTERT induction

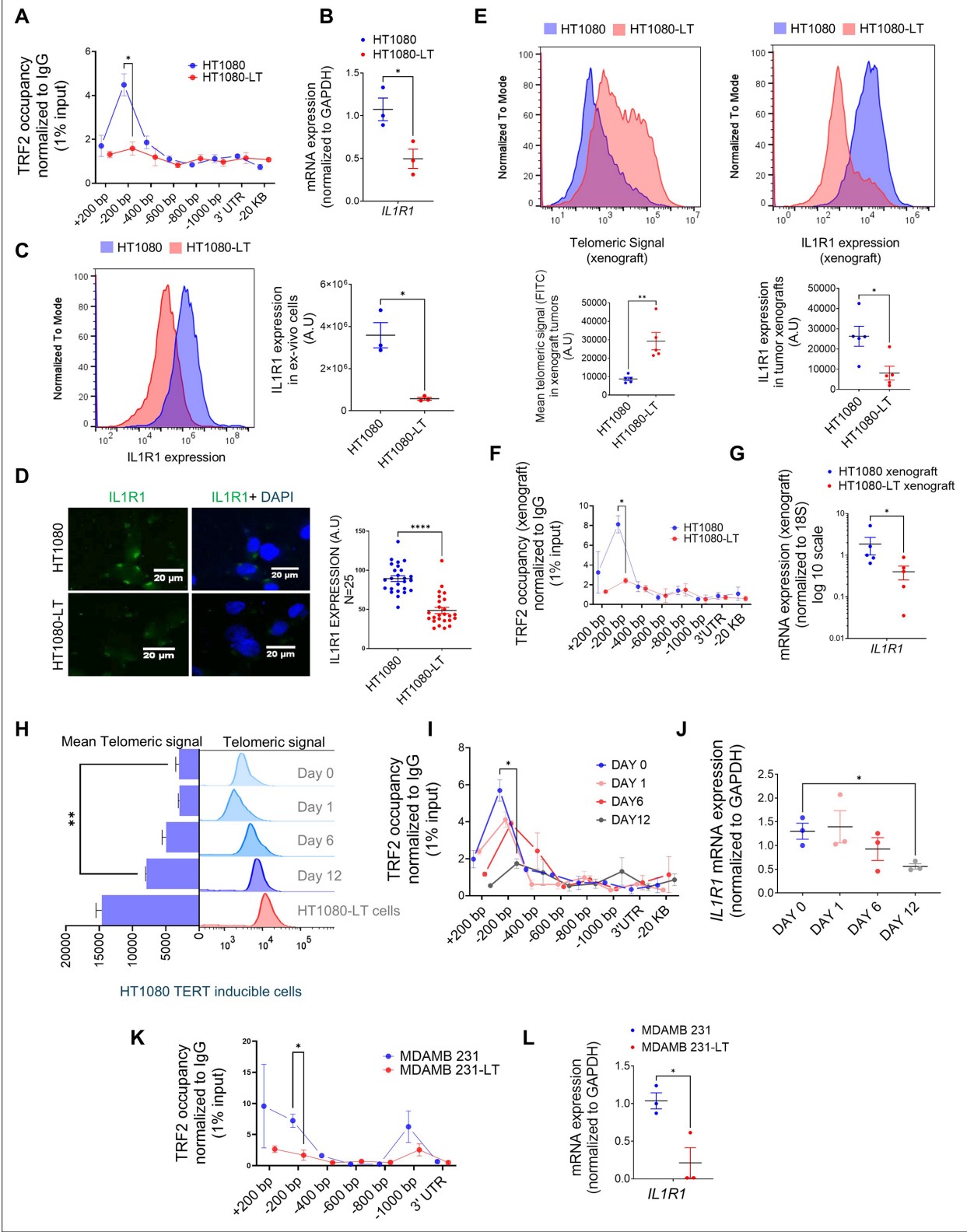

**Figure 1.** Expression of IL1R1 is regulated by telomere length-dependent enrichment of non-telomeric TRF2 on the gene promoter. (**A**) Occupancy of TRF2 was checked by ChIP-qPCR on the *IL1R1* promoter spanning +200 to –1000 bp of TSS was reduced ex vivo HT1080 cells with long telomeres (HL1080-LT) relative to ones with short telomeres (HT1080) cells (N=3 in each case); IL1R1-3'UTR or a region 20 kb upstream were used as negative controls for TRF2 binding. Statistical significance was calculated using unpaired T test with Welch's correction (p values: *≤0.05, **≤0.01, ***≤0.001,

*Figure 1 continued on next page*

*Figure 1 continued*

****≤0.0001). (**B**) mRNA expression of *IL1R1* in ex vivo HT1080 or HT1080-LT cells; *GAPDH* was used for normalisation [N=3]. Statistical significance was calculated using unpaired T test with Welch's correction (p values: *≤0.05, **≤0.01, ***≤0.001, ****≤0.0001). (**C**) IL1R1 protein expression was checked by immuno-flow cytometry in HT1080 or HT1080-LT cells in three independent replicates; Mean IL1R1 expression has been plotted along the X-axis in log scale (right panel) [N=3]. Statistical significance was calculated using unpaired T test with Welch's correction (p values: *≤0.05, **≤0.01, ***≤0.001, ****≤0.0001). (**D**) IL1R1 levels by immunofluorescence microscopy in HT1080 or HT1080-LT cells; cells were stained with DAPI for marking the cell nucleus; IL1R1 levels from 25 individual cells shown in graph (right panel). [N=25] Statistical significance was calculated using Mann-Whitney's non-parametric test (p values: *≤0.05, **≤0.01, ***≤0.001, ****≤0.0001). (**E**) Telomere length of the xenograft tumours made in NOD SCID mice using HT1080 cells with long telomeres (HL1080-LT) and short telomeres (HT1080) was checked by flow-cytometry and was higher in HT1080-LT xenograft tumours. Following this, IL1R1 expression was checked by flow cytometry for these samples. Mean Telomeric signal and Mean IL1R1 expression has been plotted respectively (panels below) [N=5] Statistical significance was calculated using Mann-Whitney's non-parametric test (p values: *≤0.05, **≤0.01, ***≤0.001, ****≤0.0001). (**F**) Occupancy of TRF2 by ChIP-qPCR at the *IL1R1* promoter spanning +200 to –1000 bp of TSS was reduced in xenograft tumours made from HT1080 cells with long telomeres (HL1080-LT) relative to ones with short telomeres (HT1080) cells (N=3 in each case; IL1R1-3'UTR or a region 20 kb upstream were used as negative controls for TRF2 binding). [N=3] Statistical significance was calculated using unpaired T test with Welch's correction (p values: *≤0.05, **≤0.01, ***≤0.001, ****≤0.0001). (**G**) mRNA expression of *IL1R1* in xenograft tumours (HT1080 or HT1080-LT cells); *18 S* was used for normalisation. [N=5] Statistical significance was calculated using Mann-Whitney's non-parametric test (p values: *≤0.05, **≤0.01, ***≤0.001, ****≤0.0001). (**H**) Telomere length by flow-cytometry of telomeric signal in HT1080 cells (hTERT-inducible stable line) following 0, 1, 6, 12, or 20 days of hTERT induction; HT1080-LT shown as positive control for enhanced telomere length; mean telomeric signal (FITC) plotted bar graph (left). [N=3] Statistical significance was calculated using unpaired T test with Welch's correction (p values: *≤0.05, **≤0.01, ***≤0.001, ****≤0.0001). (**I**) TRF2 occupancy ChIP-qPCR on the *IL1R1* promoter spanning +200 to –1000 bp of TSS in HT1080 cells following 0, 1, 6, or 12 days of hTERT induction in ex vivo culture; primers and normalisation as in **A**; IL1R1-3'UTR or a region 20 kb upstream used as negative controls. [N=3] Statistical significance was calculated using unpaired T test with Welch's correction (p values: *≤0.05, **≤0.01, ***≤0.001, ****≤0.0001). (**J**) *IL1R1* mRNA expression (normalised to *GAPDH*) in HT1080 cells following 0, 1, 6, or 12 days of hTERT induction in ex vivo culture. [N=3] Statistical significance was calculated using unpaired T test with Welch's correction (p values: *≤0.05, **≤0.01, ***≤0.001, ****≤0.0001). (**K**) TRF2 occupancy by ChIP-qPCR on the *IL1R1* promoter in MDAMB23 or MDAMD231-LT cells using primers and normalisation described earlier. [N=3] Statistical significance was calculated using unpaired T test with Welch's correction (p values: *≤0.05, **≤0.01, ***≤0.001, ****≤0.0001). (**L**) Expression for *IL1R1* in MDAMB231 cells or MDAMD231-LT cells; *GAPDH* was used for normalisation. [N=3] Statistical significance was calculated using unpaired T test with Welch's correction (p values: *≤0.05, **≤0.01, ***≤0.001, ****≤0.0001). Error bars correspond to SEM from independent experiments.

The online version of this article includes the following source data and figure supplement(s) for figure 1:

**Source data 1.** Source data for all plots in *Figure 1* and corresponding details of statistical tests and p-values.

**Figure supplement 1.** Graphical representation of the Telomere Sequestration Partitioning (TSP) model.

**Figure supplement 2.** Expression of IL1R1 is regulated by telomere length-dependent enrichment of non-telomeric TRF2 on the gene promoter (continued).

**Figure supplement 2—source data 1.** Source data for all plots in *Figure 1—figure supplement 2* and corresponding details of statistical tests and p-values.

**Figure supplement 2—source data 2.** PDF file containing original western blot for *Figure 1—figure supplement 2I* indicating the relevant bands and treatments.

**Figure supplement 2—source data 3.** Original image files for western blot for *Figure 1—figure supplement 2I*.

(*Figure 1H*, *Figure 1—figure supplement 2K*): gradual reduction of *IL1R1* promoter TRF2 binding (*Figure 1I*) and *IL1R1* expression (*Figure 1J*) with telomere elongation over the 12 days was notably consistent with observations made using HT1080/HT1080-LT cells (*Figure 1A and B*).

### Telomere elongation model in MDAMB231 breast cancer cell line

An alternative telomere elongation model was made from TNBC MDAMB-231 cells using G-rich-telomere-repeat (GTR) oligonucleotides as reported earlier (*Mukherjee et al., 2018*; *Wright et al., 1996*). In the resulting MDAMB-231-LT cells with ~1.8- to twofold increase in TL (*Figure 1—figure supplement 2L*), *IL1R1* promoter TRF2 occupancy and *IL1R1* expression were significantly lower (*Figure 1K and L*). The results collectively suggest that TRF2 occupancy at the *IL1R1* promoter and the expression of the gene is telomere length dependent.

### TRF2 transcriptionally activates *IL1R1*

We, next, investigated the role of TRF2 in the regulation of *IL1R1*. TRF2 over-expression (Flag-TRF2) showed an increase in *IL1R1* mRNA expression and protein, and TRF2 silencing reduced *IL1R1* in HT1080 cells (*Figure 2A*, *Figure 2—figure supplement 1A*) indicating that TRF2 activated *IL1R1* expression. Flag-tagged TRF2-full-length, or a truncated version of flag-tagged TRF2 protein without

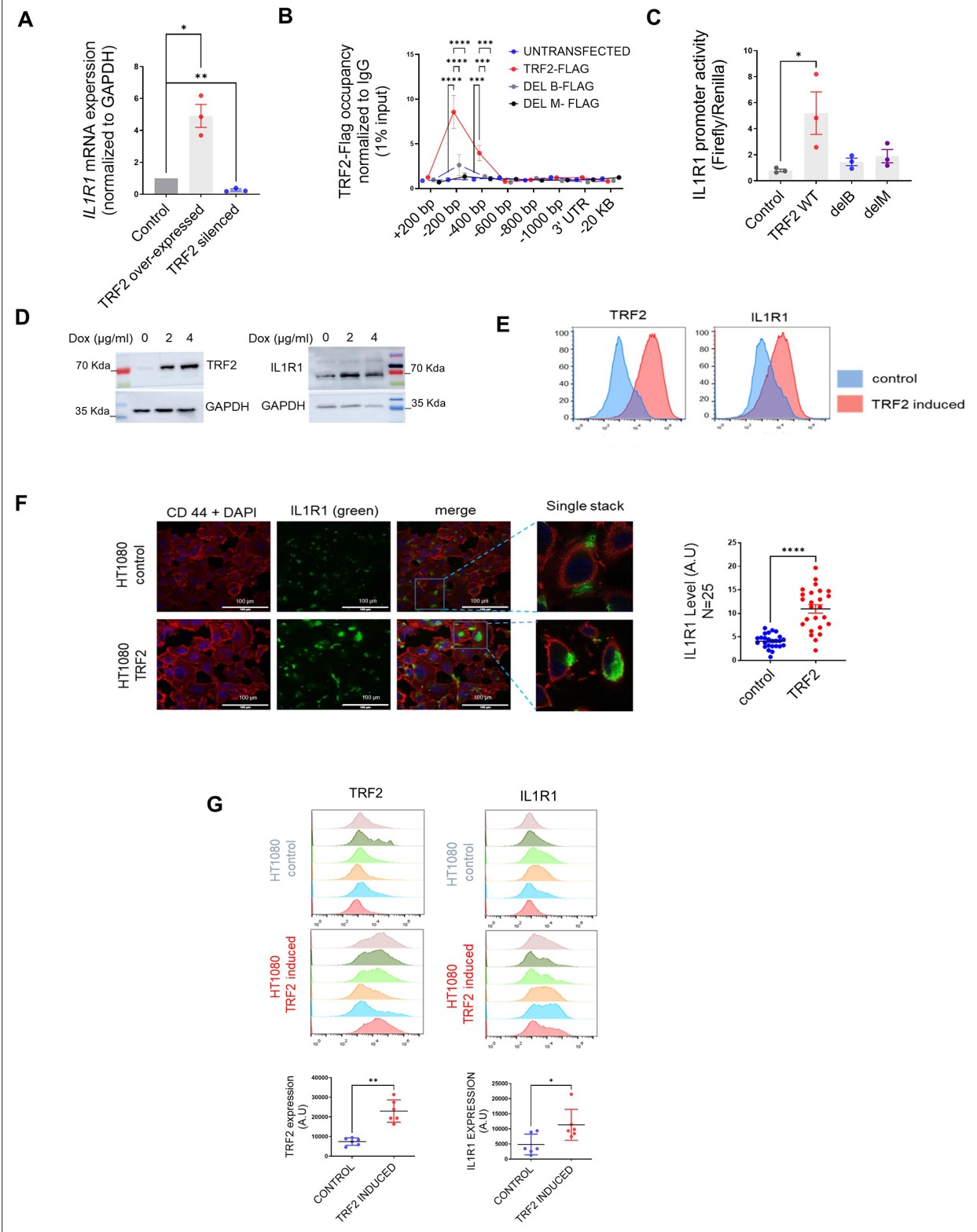

**Figure 2.** TRF2 is a transcriptional activator of IL1R1. (**A**) *IL1R1* expression by qRT PCR following expression of TRF2-flag or TRF2 silencing (48 hr of transient transfection); *GAPDH* was used for normalisation. [N=3] Statistical significance was calculated using unpaired T test with Welch's correction (p values: *≤0.05, **≤0.01, ***≤0.001, ****≤0.0001). (**B–C**) TRF2 ChIP-qPCR spanning +200 to –1000 bp of *IL1R1* TSS for TRF2 occupancy (**B**) and promoter activity (luciferase reporter including –1500 bp TSS); (**C**) following expression of flag-tag TRF2 full-length, or the mutants TRF2-delB or TRF2-delM; anti-flag antibody used for ChIP; IL1R1-3'UTR or a region 20 kb upstream were used as negative controls. [N=3] Statistical significance was calculated

*Figure 2 continued on next page*

*Figure 2 continued*

using Tukey's multiple comparisons test (p values: *≤0.05, **≤0.01, ***≤0.001, ****≤0.0001). (**D**) TRF2 and IL1R1 levels by western blots following TRF2 induction with 2 or 4 µg/ml doxycycline (Dox) in HT1080 cells (stably transformed for Dox-inducible TRF2); GAPDH as loading. (**E**) TRF2 and IL1R1 expression in control or TRF2-induced HT1080 cells by flow-cytometry; TRF2/IL1R1 in x-axis in log scale for 20,000 cells. (**F**) IL1R1 levels in control or TRF2-induced HT1080 cells. CD44 used for cell-surface marker and nuclei were stained with DAPI; 25 cells in each condition were scored and plotted in the summary graph. [N=25] Statistical significance was calculated using Mann-Whitney's non-parametric test (p values: *≤0.05, **≤0.01, ***≤0.001, ****≤0.0001). (**G**) TRF2 (left panel), IL1R1 (right panel) levels from xenograft tumours in NOD-SCID mice (control or doxycycline-induced TRF2 in HT1080 cells; N=6 mice in each group) by immuno-flow cytometry; mean fluorescence signal from individual tumours in control or TRF2-induced tumours plotted in adjacent graphs. [N=6] Statistical significance was calculated using Mann-Whitney's non-parametric test (p values: *≤0.05, **≤0.01, ***≤0.001, ****≤0.0001). Error bars correspond to SEM from independent experiments.

The online version of this article includes the following source data and figure supplement(s) for figure 2:

**Source data 1.** Source data for all plots in *Figure 2* and corresponding details of statistical tests and p-values.

**Source data 2.** PDF file containing original western blot for *Figure 2D* indicating the relevant bands and treatments.

**Source data 3.** Original image files for western blot or *Figure 2D*.

**Figure supplement 1.** TRF2 is a transcriptional activator of IL1R1 (continued).

**Figure supplement 1—source data 1.** Source data for all plots in *Figure 2—figure supplement 1* and corresponding details of statistical tests and p-values.

**Figure supplement 1—source data 2.** PDF file containing original western blot for *Figure 2—figure supplement 1A and B* indicating the relevant bands and treatments.

**Figure supplement 1—source data 3.** Original image files for western blot for *Figure 2—figure supplement 1A and B*.

the necessary domains for DNA binding (N-terminal Basic-domain [TRF2-delB] or C terminal MYB-domain TRF2-delM), showed that both TRF2 occupancy and *IL1R1* promoter activity was compromised in case of the mutants; indicating TRF2 DNA binding as necessary for *IL1R1* activation (*Figure 2B and C*). We developed a doxycycline-inducible TRF2 model using HT1080 cells. Upon induction of TRF2, IL1R1 expression was also enhanced (*Figure 2D and E*). We checked and found that IL1R1 expression of the cell surface was also significantly higher in cells with higher TRF2 (*Figure 2F*). Further, TRF2-dependent activation of *IL1R1* was clear in MDAMB-231 cells (*Figure 2—figure supplement 1B*), normal MRC5 fibroblasts and HEK293T cells (*Figure 2—figure supplement 1C*). HT1080 cells with doxycycline-inducible-TRF2 were used to make tumour xenografts in NOD-SCID mice. TRF2-induced tumours had enhanced IL1R1 relative to the xenograft tumours where TRF2 was not induced (*Figure 2G*).

## G-quadruplex DNA secondary structure is necessary for TRF2 binding to the *IL1R1* promoter

We reported TRF2 binding to G-quadruplex (G4) DNA secondary structures within promoters throughout the genome *Mukherjee et al., 2019a*; further, the role of G4s in epigenetic regulation was noted (*Mukherjee et al., 2019b*). Here, we found two G4s within the *IL1R1* promoter TRF2-binding site (*Figure 3A*, *Figure 3—figure supplement 1A*). We confirmed that the DNA sequences formed G4 structures in solution and specific base substitutions disrupted the structure formation (*Figure 3B*); both G4 forming motif sequences showed the characteristic Circular Dichroism signatures for 'parallel' G-quadruplex structures which were largely abrogated when the sequences were mutated (*Figure 3B*). BG4 antibody [against G4s (*Biffi et al., 2013*)] ChIP was enriched ~200 bp upstream on the *IL1R1* promoter in HT1080 cells (*Figure 3C*); TRF2-dependent *IL1R1* promoter activity was significantly compromised on introducing G4-deforming G>T substitutions on both the G4 motifs (*Figure 3D*), while the effect of the mutation was more for G4 motif B. In the presence of the intracellular G4-binding ligand 360 A (*Granotier et al., 2005*), the TRF2 occupancy in HT1080 cells was found to be significantly reduced at the *IL1R1* promoter consistent with results from other TRF2 binding sites (*Mukherjee et al., 2019a*; *Figure 3—figure supplement 1B*). Upon treatment of HT1080 cells with 360 A, TRF2-induced IL1R1 promoter activity, and expression/protein were significantly reduced (*Figure 3E*). Using CRISPR, we artificially inserted an *IL1R1* luciferase-reporter-cassette (–1500 bp) at the CCR5 safe-harbour locus in HEK293T cells; TRF2-dependent promoter activity from the inserted *IL1R1* promoter-reporter and TRF2 binding was reduced when the G4 motif B was mutated supporting earlier results (*Figure 3F and G*).

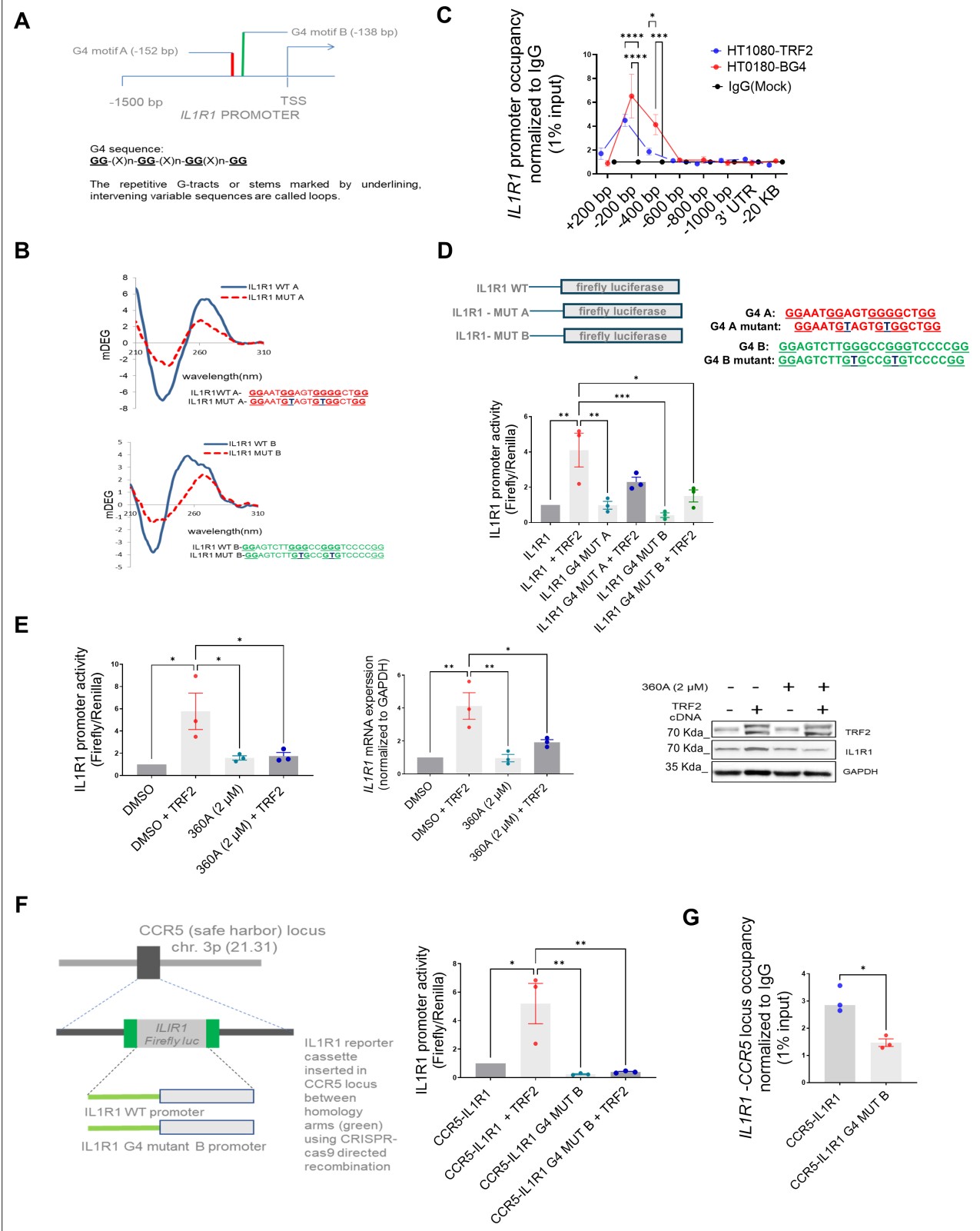

**Figure 3.** DNA secondary structure G-quadruplexes are important for TRF2 binding to IL1R1 promoter. (**A**) Schematic depicting G4 motifs (**A, B**) and their respective positions on the *IL1R1* promoter; sequence scheme depicting generic G4 motifs; (**B**) Circular dichroism profile (220–310 nm wavelength) of oligonucleotides (5 μM) with sequence of the G4 motifs A or B (or respective mutants with base substitutions) in solution. (**C**) ChIP-qPCR following BG4 ChIP at the IL1R1 promoter (fold enrichment over mock) overlaid with TRF2 occupancy (fold enrichment over IgG) in HT1080 cells. [N=3] Statistical

*Figure 3 continued*

significance was calculated using Šídák's multiple comparisons test (p values: *≤0.05, **≤0.01, ***≤0.001, ****≤0.0001). (**D**) IL1R1 promoter activity in HT1080 cells from luciferase-reporter without or with G4-deforming substitutions against G4 motif A (IL1R1G4-MUT-A) or G4 motif B (IL1R1-G4 MUT-B) in presence or absence of TRF2 induction; G4 motifs A and B as shown in schematic; specific base substitutions have been indicated in blue font. [N=3] Statistical significance was calculated using Tukey's multiple comparisons test (p values: *≤0.05, **≤0.01, ***≤0.001, ****≤0.0001). (**E**) *IL1R1* promoter activity, mRNA and protein levels in presence/absence of the G4 binding ligand 360 A with or without TRF2 induction. For Left and centre panels- [N=3] Statistical significance was calculated using Tukey's multiple comparisons test (p values: *≤0.05, **≤0.01, ***≤0.001, ****≤0.0001). (**F**) IL1R1 promoter-firefly luciferase reporter cassette, with or without substitutions deforming the G4 motif B, was artificially inserted at the CCR5 locus using CRISPR-cas9 gene editing in HEK293T cells (scheme in left panel). Promoter activity from (right panel). [N=3] Statistical significance was calculated using Tukey's multiple comparisons test (p values: *≤0.05, **≤0.01, ***≤0.001, ****≤0.0001). (**G**) TRF2 occupancy by ChIP-qPCR (right panel) at the artificially inserted IL1R1 promoter without or with G4-deforming substitutions (IL1R1G4 MUT-B);ChIP-qPCR primers designed specific to the inserted promoter using the homology arms(see Materials and methods). [N=3] Statistical significance was calculated using unpaired T test with Welch's correction (p values: *≤0.05, **≤0.01, ***≤0.001, ****≤0.0001). Error bars correspond to SEM from independent experiments.

The online version of this article includes the following source data and figure supplement(s) for figure 3:

**Source data 1.** Source data for all plots in *Figure 3* and corresponding details of statistical tests and p-values.

**Source data 2.** PDF file containing original western blot for *Figure 3E* indicating the relevant bands and treatments.

**Source data 3.** Original image files for western blot for *Figure 3E*.

**Figure supplement 1.** DNA secondary structure G-quadruplexes are important for TRF2 binding to IL1R1 promoter (continued).

**Figure supplement 1—source data 1.** Source data for all plots in *Figure 3—figure supplement 1* and corresponding details of statistical tests and p-values.

Furthermore, CRISPR-mediated G>T editing in the G4 motif (motif B) within the endogenous *IL1R1* promoter resulted in a reduction of TRF2 binding and *IL1R1* expression (*Figure 3—figure supplement 1C and D*).

## TRF2-dependent recruitment of histone acetyltransferase p300 induces H3K27ac modification at the *IL1R1* promoter

We had previously reported that TRF2 occupancy alters the local epigenetics at gene promoters (*Mukherjee et al., 2018*). Here, we sought to understand if TRF2 occupancy brought about any epigenetic alteration at the *IL1R1* promoter. Screening of activation (H3K4me3 and H3K27ac) and repressive (H3K27me3, H3K9me3) histone marks at the *IL1R1* promoter following TRF2 induction, showed enhanced H3K27acetylation (*Figure 4—figure supplement 1A*). In both HT1080 and MDAMB-231 cells, *IL1R1* promoter H3K27 acetylation increased/reduced on TRF2 induction/silencing respectively (*Figure 4A*). We sought to find a potential histone acetyltransferase (HAT) and noted canonical HATs p300/CBP (*Roth et al., 2001*) at the *IL1R1* promoter in multiple cell lines in ENCODE (*Figure 4—figure supplement 1B*). ChIP-qPCR confirmed p300 occupancy within upstream 200–400 bp promoter of *IL1R1* in TRF2-up conditions, which reduced significantly on TRF2 downregulation, in HT1080 and MDAMB-231 cells (*Figure 4B*). Furthermore, occupancy of acetylated-p300/CBP (activated HAT) was enriched at the *IL1R1* promoter on TRF2-induction in HT1080 cells (*Figure 4C*). Together these showed TRF2-mediated recruitment and acetylation of p300/CBP, resulting in H3K27 acetylation at the *IL1R1* promoter. *IL1R1* promoter occupancy of p300, CBP, ac-p300/CBP and H3K27ac was relatively low in HT1080-LT compared to HT1080 cells with shorter telomeres (*Figure 4D*) consistent with lower TRF2 occupancy at the *IL1R1* promoter in LT cells (*Figure 1A*).

A recent work showed that TRF2 can physically interact with p300 and p300 acetylates TRF2 at specific lysine residues (*Her and Chung, 2013*). We found that TRF2 directly interacted with p300, evident from co-immunoprecipitation using lysate from HT1080 cells (*Figure 4—figure supplement 1C*). To ascertain direct TRF2-p300 interaction and consequent H3K27 acetylation, we did histone-acetyl-transfer assays using purified p300, histone H3, and acetyl-CoA (substrate for the acetyl group) with or without recombinant TRF2. Transfer of the acetyl group to histone H3 was significantly enhanced in the presence of both TRF2 and p300, relative to p300 only; and TRF2 co-incubated with p300 without H3 gave marginal HAT activity (*Figure 4—figure supplement 1D*). These together demonstrate a direct function of TRF2 in recruitment and HAT activity of p300 in H3K27 acetylation at the *IL1R1* promoter.

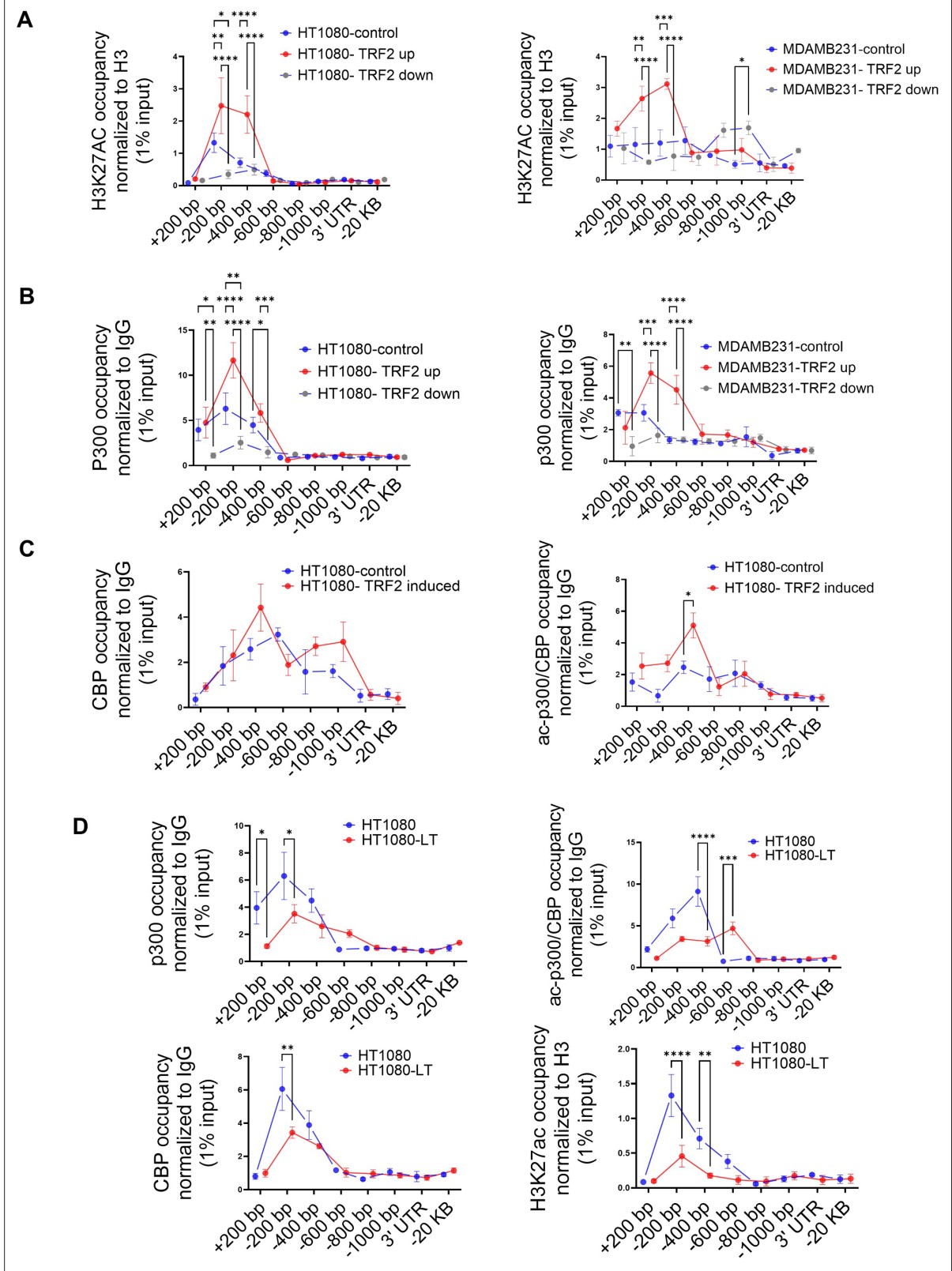

**Figure 4.** TRF2 recruits the histone acetyl transferase p300 to the *IL1R1* promoter. (**A**) H3K27ac occupancy at the *IL1R1* promoter spanning +200 to −1000 bp of TSS by ChIP-qPCR in HT1080 (left panel) and MDAMB231 cells (right panel) in control (uninduced), TRF2-induced (up) or TRF2-down conditions; IL1R1-3'UTR or a region 20 kb upstream were used as negative controls. [N=3] Statistical significance was calculated using Tukey's multiple comparisons test (p values: *≤0.05, **≤0.01, ***≤0.001, ****≤0.0001). (**B**) p300 occupancy on the *IL1R1* promoter in HT1080 (left panel) and MDAMB231

*Figure 4 continued on next page*

Figure 4 continued

cells (right panel) in control (uninduced), TRF2-induced or TRF2-down conditions; IL1R1-3'UTR or a region 20 kb upstream were used as negative controls. [N=3] Statistical significance was calculated using Tukey's multiple comparisons test (p values: *≤0.05, **≤0.01, ***≤0.001, ****≤0.0001). (**C**) CBP (left panel) and ac-p300/CBP (right panel) occupancy on the *IL1R1* promoter in HT1080 cells in control (uninduced) or TRF2-induced conditions; IL1R1-3'UTR or a region 20 kb upstream were used as negative controls. [N=3] Statistical significance was calculated using Šídák's multiple comparisons test (p values: *≤0.05, **≤0.01, ***≤0.001, ****≤0.0001). (**D**) p300, CBP, acp300/CBP and H3K27Ac occupancy on the *IL1R1* promoter in HT1080 and HT1080-LT cells; IL1R1-3'UTR or a region 20 kb upstream were used as negative controls. [N=3] Statistical significance was calculated using Šídák's multiple comparisons test (p values: *≤0.05, **≤0.01, ***≤0.001, ****≤0.0001). All protein ChIP, other than histones, were normalised to 1% input and fold-change have been calculated over respective IgG. Histone ChIP were normalised to 1% Input and fold change over total H3 have been calculated for individual samples (Further details in the Materials and methods section). Error bars correspond to SEM from independent experiments.

The online version of this article includes the following source data and figure supplement(s) for figure 4:

**Source data 1.** Source data for all plots in *Figure 4* and corresponding details of statistical tests and p-values.

**Figure supplement 1.** TRF2 recruits the histone acetyl transferase p300 to the IL1R1 promoter (continued).

**Figure supplement 1—source data 1.** Source data for all plots in *Figure 4—figure supplement 1* and corresponding details of statistical tests and p-values.

**Figure supplement 1—source data 2.** PDF file containing original western blot for *Figure 4—figure supplement 1C* indicating the relevant bands and treatments.

**Figure supplement 1—source data 3.** Original image files for western blot for *Figure 4—figure supplement 1C*.

## Acetylation of TRF2 at Lysine residue 293 is necessary for p300 recruitment on the *IL1R1* promoter

We sought to understand if the acetylation of TRF2 was important for p300-TRF2 complex recruitment to the *IL1R1* promoter. We found that several residues on TRF2 have been reported to be acetylated / deacetylated (*Rizzo et al., 2017*; *Her and Chung, 2013*; *Figure 5A*). To test the direct function of TRF2 acetylation in p300 recruitment, Flag-TRF2 mutants [K176R, K179R, K190R, and K293R (*Rizzo et al., 2017*; *Her and Chung, 2013*)] were screened (*Figure 5B*) for effect on *IL1R1* expression. TRF2-K293R gave notably compromised activation of *IL1R1* at mRNA and protein levels (*Figure 5B–D*). The binding of flag-TRF2-293R at the *IL1R1* promoter was similar to TRF2-WT (*Figure 5E*) suggesting that the loss in *IL1R1* activation was not due to lower promoter binding by TRF2-293R. However, TRF2-293R significantly reduced p300 and ac-p300/CBP recruitment on the *IL1R1* promoter relative to TRF2-wildtype (WT) induction (*Figure 5E*). TRF2 293R also showed lower physical interaction with p300 in an immunoprecipitation experiment in comparison to TRF2 WT (*Figure 5—figure supplement 1A*). The TRF2 mutant K293R, devoid of H3K27 acetylation activity, gave a loss of function (*Figure 5B–D*). Therefore, TRF2-K293R was fused to dead-Cas9 (dCas9-TRF2-K293R), and dCas9-TRF2 was used as control. On using sgRNA against the *IL1R1* promoter, we found significant activation or downregulation of *IL1R1* in the case of dCas9-TRF2 or dCas9-TRF2-K293R, respectively, in HT1080 and MDAMB231 cells. In contrast, other TRF2-target genes (*Mukherjee et al., 2018*) remained unaffected indicating single-gene specificity (*Figure 5F*). We further checked the role of the TRF2 mutant K293R on other targets of TRF2 that we had previously validated (*Mukherjee et al., 2019a*; *Figure 5—figure supplement 1B*). While the TRF2 mutant K293R did not show much effect on genes wherein TRF2 acted as a transcriptional repressor, the effect on the candidate 'activation' targets of TRF2 was not uniform as well. The Lysine 293 in TRF2 seems to be important for activation but not repression. However, while the mutant K293R diminished TRF2-mediated activation of *IL1R1* and *PDGFR-B*, it did not affect *WRNIP1*. This suggests that the TRF2 PTMs might have context-dependence at specific loci in the genome.

## Cancer cells with higher TRF2 are sensitive to IL1-mediated NF-kappaB (p65) activation

Primarily ligands IL1A/B via receptors IL1R1/R2 activate IL1 signalling through NF-KappaB (p65/RELA)-Ser536 phosphorylation (*Weber et al., 2010*). This induces *IL1B* and other inflammatory genes including *IL6, IL8*, and *TNF*. Here, we checked NF-kappaB-Ser536 phosphorylation following stimulation with IL1B in the presence/absence of TRF2 in HT1080 and MDAMB-231 cells. TRF2-induction gave relatively enhanced NF-kappaB-Ser536 phosphorylation that sustained for a longer duration compared to uninduced controls in both cell types; total NF-kappaB levels remained unaltered (*Figure 6A and B*). Accordingly, on stimulation with either IL1A or IL1B in a TRF2-silenced condition,

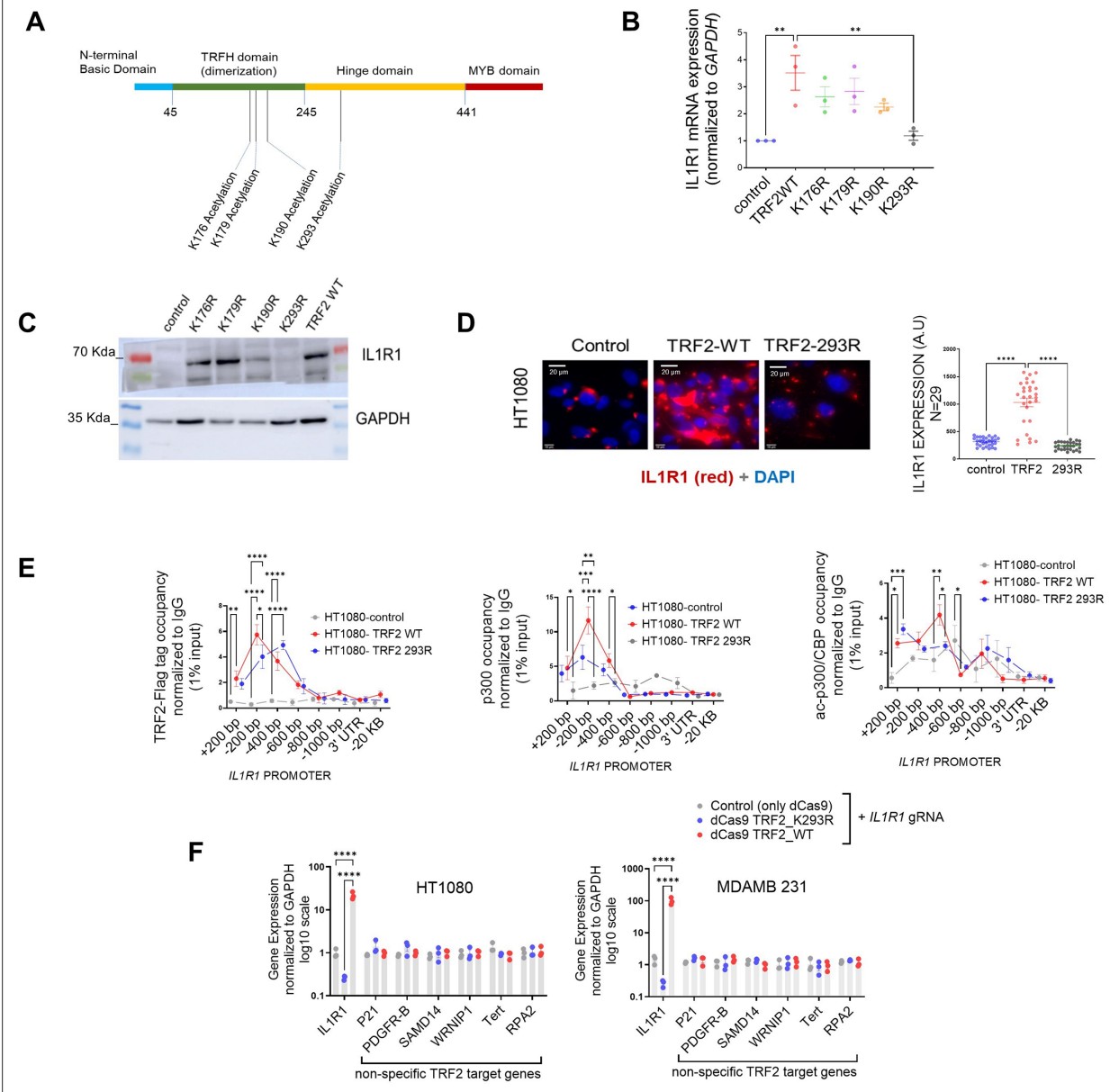

**Figure 5.** Acetylation of TRF2 at Lysine residue 293 is necessary for p300 recruitment on the *IL1R1* promoter. (**A**) Schematic showing positions (amino acid residues) where acetylation and deacetylation of TRF2 have been reported. (**B**) IL1R1 mRNA expression post 48 hr of transient over-expression of various TRF2 acetylation mutants. *GAPDH* was used for normalisation. [N=3] Statistical significance was calculated using Dunnett's multiple comparisons test (p values: *≤0.05, **≤0.01, ***≤0.001, ****≤0.0001). (**C**) IL1R1 protein level following post 48 hr of transient over-expression of various TRF2 acetylation mutants; GAPDH used as loading control. (**D**) Immunofluorescence for IL1R1 in HT1080 cells without (control) or with induction of flag-tag TRF2 WT or mutant TRF2-293R; quantification from 29 cells in each case shown in graph. [N=29] Statistical significance was calculated using Mann-Whitney's non-parametric test (p values: *≤0.05, **≤0.01, ***≤0.001, ****≤0.0001). (**E**) Occupancy of flag-tag TRF2-WT or TRF2-293R on the IL1R1 promoter by ChIP-qPCR in HT1080 cells (left panel); IL1R1-3'UTR or a region 20 kb upstream were used as negative controls. Occupancy by ChIP-qPCR of p300 (middle panel) and acp300/CBP (right panel) without (control) or with induction of TRF2-WT and TRF2-293R. [N=3] Statistical significance was calculated using Tukey's multiple comparisons test (p values: *≤0.05, **≤0.01, ***≤0.001, ****≤0.0001). (**F**) TRF2-wild type (WT) and TRF2-2293R mutant proteins fused with dCAS9 expressed and targeted to the IL1R1 promoter using *IL1R1*-specific gRNA in HT1080 and MDAMB231 cells. Following this, expression of *IL1R1* or other TRF2 target genes (non-specific with respect to the IL1R1-gRNA) in HT1080 (left) or MDMB231 cells (right). [N=3] Statistical significance was calculated using Tukey's multiple comparisons test (p values: *≤0.05, **≤0.01, ***≤0.001, ****≤0.0001). Error bars correspond to SEM from independent experiments.

The online version of this article includes the following source data and figure supplement(s) for figure 5:

**Source data 1.** Source data for all plots in *Figure 5* and corresponding details of statistical tests and p-values.

*Figure 5 continued on next page*

*Figure 5 continued*

**Source data 2.** PDF file containing original western blot for *Figure 5C* indicating the relevant bands and treatments.

**Source data 3.** Original image files for western blot for *Figure 5C*.

**Figure supplement 1.** Acetylation of TRF2 at Lysine residue 293 is necessary for p300 recruitment on the IL1R1 promoter (continued).

**Figure supplement 1—source data 1.** Source data for all plots in *Figure 5—figure supplement 1* and corresponding details of statistical tests and p-values.

**Figure supplement 1—source data 2.** PDF file containing original western blot for *Figure 5—figure supplement 1A* indicating the relevant bands and treatments.

**Figure supplement 1—source data 3.** Original image files for western blot for *Figure 5—figure supplement 1A*.

expression of the NF-kappaB targets *IL6, IL8,* and *TNF* were reduced (*Figure 6C*). Further, the ligand TNFα was reported to induce NF-kappaB in an IL1-independent fashion (*Liu et al., 2017*) suggesting this pathway would be unaffected by the presence/absence of TRF2. Indeed, the expression of NF-kappaB targets (*IL6, IL8,* and *TNF*) remained unaffected upon stimulation by TNFαin the absence of TRF2, supporting the role of TRF2 in IL1 signalling specifically (*Figure 6C*). Additionally, TRF2 induction enhanced the expression of *IL6*, *IL8,* and *TNF*, which was rescued in the presence of the receptor antagonist IL1RA, further confirming the specific role of TRF2-IL1R1 in NF-kappaB activation (*Figure 6D*). When tested in NOD SCID mouse xenograft tumour samples, we observed that the ratio of phosphor-NF-kappaB to total NF-kappaB B was significantly higher in TRF2-induced tumours (*Figure 6E*).

As shown in *Figure 5F* (mRNA expression of *IL1R1*), dCas9-TRF2-K293R diminished IL1R1 levels (*Figure 6—figure supplement 1A*).This resulted in reduced NF-kappaB-Ser536 phosphorylation indicating attenuated NF-kappaB activation in HT1080 and MDAMB 231 cells; dCas9-TRF2 as expected showed activation of IL1R1 and NF-kappaB (*Figure 6—figure supplement 1A*).

To directly test the function of IL1R1 in TRF2-mediated NF-kappaB activation, *IL1R1*-knockout (KO; CRISPR-mediated) was made in HT1080 cells having doxycline-inducible-TRF2: NF-kappaB activation (Ser536-phosphorylation), including upregulation of the NF-kappaB-target genes *IL2*, *IL6*, and *TNF*, was significantly reduced following TRF2 induction in the KO cells compared to HT1080 cells (*Figure 6—figure supplement 1B*). Together, these demonstrate the direct role of TRF2 in the induction of IL1 signalling through upregulation of *IL1R1*.

We reasoned that in cells with longer telomeres IL1 signalling would be affected due to reduced TRF2 binding at the *IL1R1* promoter resulting in attenuated activation of *IL1R1*. In the case of HT1080-LT cells, notable increase in NF-kappaB-Ser536 phosphorylation, following IL1B stimulation, was notably delayed (3 hr in HT10180-LT vs 30 min in HT1080 respectively; *Figure 6—figure supplement 1C*).

## IL1R1 expression in TNBC is sensitive to variation in telomere length

To check if our findings were clinically relevant, we used TNBC tumour samples to characterise telomere length and *IL1R1* expression. The IL1 (interleukin 1) pathway is relevant in TNBC prognosis as indicated by past studies (*Lappano et al., 2020*; *Jeon et al., 2016*), and therefore it was relevant for us to use TNBC samples. We checked 94 TNBC samples and variation in telomeres across tumours was evident (*Figure 7A–C*).

We measured TL (telomere length) in tissues from 94 breast cancer (TNBC) patients: Median TL was ~3.6 kb (quantitative-telomeric-RTPCR; *O'Callaghan et al., 2008*; Telo-qPCR-see Materials and methods; *Figure 7A*, *Supplementary file 3*). Tumours with TL <50% of median were designated TNBC-short-telomeres (TNBC-ST) and >50% of median length TNBC-long-telomeres (TNBC-LT; *Figure 1D*). TLs were confirmed by flow cytometry and in situ hybridization (*Figure 7B*).

Further, reads from the whole genome sequence of the TNBC samples were analyzed for estimation of TL using the reported pipline- TelSeq (*Ding et al., 2014*): TL difference between TNBC-ST and LT was clear, and cancer tissue had lower TL than adjacent normal tissue consistent with earlier reports (*Aviv et al., 2017*). We found that, while there were inter-tumour telomere length variations, about ~70% or more (45 out of 66 TNBC samples and 11 out of 14 in tel-seq analysis) samples for which data was available for adjacent normal tissue – showed lower telomere length in tumour samples (*Figure 7D*).

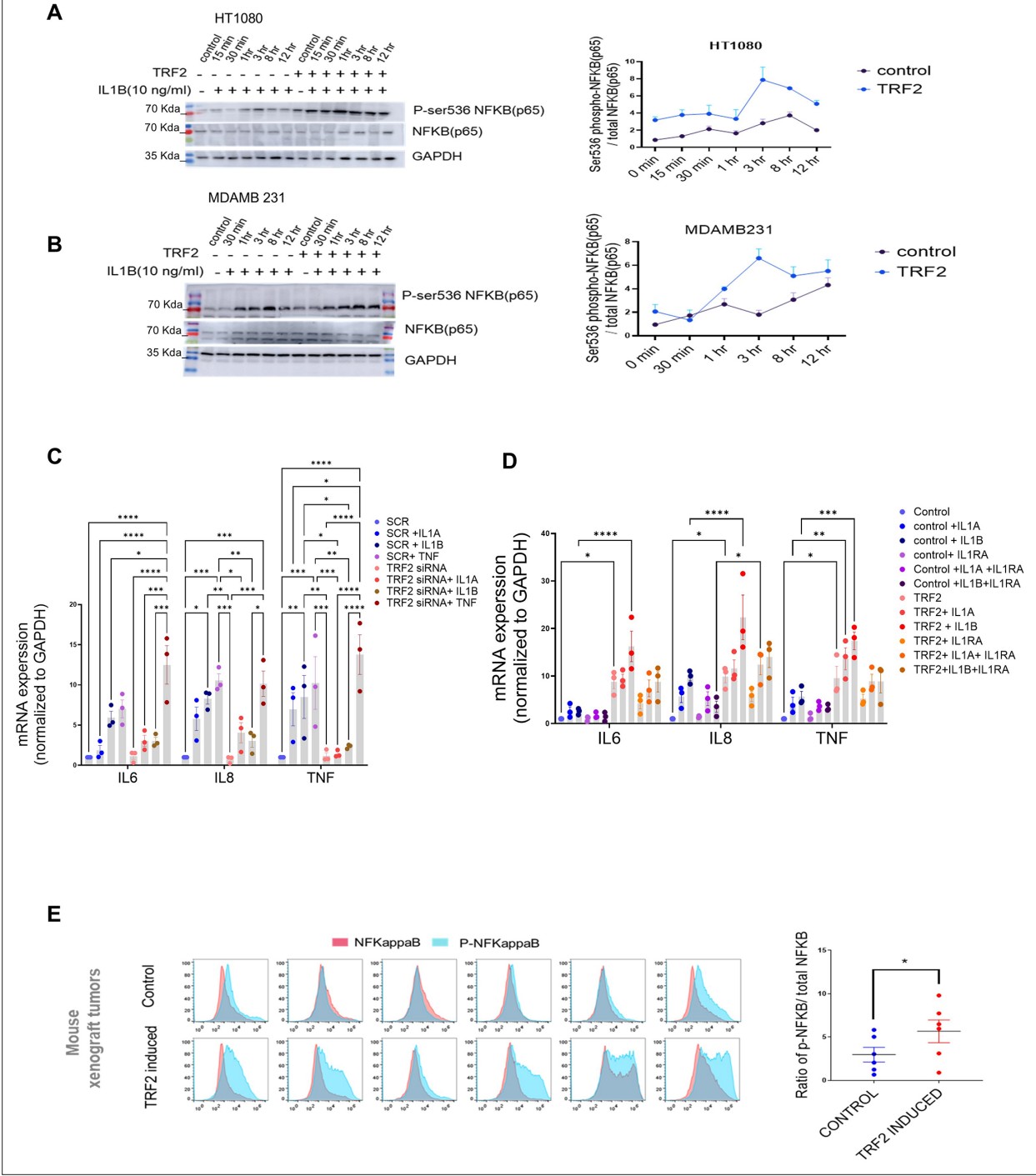

**Figure 6.** TRF2-dependent IL1 pathway activates NFkappa B (p65) in cancer cells. (**A, B**) NFKappaB activation in presence of IL1B (10 ng/ml) in HT1080 (**A**) or MDAMB231 (**B**) cells with or without TRF2 induction; activation signalling was confirmed through NFKappaB-Ser536 phosphorylation (normalised to total NFKappaB); ratio of Ser566-p/total NFKappaB plotted for respective blots from three independent replicates (right panels). [N=2]. (**C**) Expression of NFKappaB targets *IL6, IL8* or *TNF* in presence/absence IL1A, IL1B, or TNF-a (10 ng/ml) for 24 hr in control (scrambled siRNA) or TRF2-low (TRF2 siRNA) conditions in HT1080 cells. [N=3] Statistical significance was calculated using Tukey's multiple comparisons test (p values: *≤0.05, **≤0.01, ***≤0.001, ****≤0.0001). (**D**) Expression of *IL6, IL8,* or *TNF* in control or TRF2-induced conditions on treatment with either IL1A or IL1B (10 ng/ml) for 24 hr in absence (left panel) or presence (right panel) of the IL1-receptor-antagonist IL1RA (20 ng/ml) in HT1080 cells. [N=3] Statistical significance was calculated using Tukey's multiple comparisons test (p values: *≤0.05, **≤0.01, ***≤0.001, ****≤0.0001). (**E**) NFkappaB/phosphor-Ser536-NFKappaB levels in xenograft tumours developed in NOD-SCID mice (control or doxycycline-induced TRF2 in HT1080 cells; N=6 mice in each group) by immuno-flow cytometry; mean fluorescence signal from individual tumours in control or TRF2-induced tumours plotted in adjacent graph; activation shown as

*Figure 6 continued on next page*

*Figure 6 continued*

ratio of pSer536-p-NFkappaB over total NFkappaB; significance was calculated using Wilcoxon's non-parametric test. [N=6] Statistical significance was calculated using Mann-Whitney's non-parametric test (p values: *≤0.05, **≤0.01, ***≤0.001, ****≤0.0001). Error bars correspond to SEM from independent experiments.

The online version of this article includes the following source data and figure supplement(s) for figure 6:

**Source data 1.** Source data for all plots in *Figure 6* and corresponding details of statistical tests and p-values.

**Source data 2.** PDF file containing original western blot for *Figure 6A and B* indicating the relevant bands and treatments.

**Source data 3.** Original image files for western blot for *Figure 6A and B*.

**Figure supplement 1.** TRF2-dependent IL1 pathway activates NFkappa B (p65) in cancer cells (continued).

**Figure supplement 1—source data 1.** Source data for all plots in *Figure 6—figure supplement 1* and corresponding details of statistical tests and p-values.

**Figure supplement 1—source data 2.** PDF file containing original western blot for *Figure 6—figure supplement 1B and C* indicating the relevant bands and treatments.

**Figure supplement 1—source data 3.** Original image files for western blot for *Figure 6—figure supplement 1B and C*.

EpCAM$^{+ve}$ (tumour-cell marker; *Went et al., 2004*) cells within the TME had significant heterogeneity in TL (lower in TNBC-ST vis-a-vis LT), whereas the difference in TL of EpCAM$^{-ve}$ cells was insignificant indicating TL variation in TME was primarily from cancer cells (*Figure 7E*, *Figure 7—figure supplement 1A*). Details for the Tel-seq data have been tabulated with details of TL estimation for individual samples (*Figure 7—figure supplement 1B*). We surprisingly found that telomerase activity negatively correlated with telomere length in TNBC (*Figure 7—figure supplement 1C*), and while *TERT* was not found to be significantly higher, *TERC* was found to be significantly enhances in TNBC-LT samples (*Figure 7—figure supplement 1D*). This indicates that *TERC* is possibly a more important determinant for telomere length in TNBC.

Next, along with *IL1R1*, a curated set of pro- and anti-inflammatory cytokines commonly studied concerning tumour immunity (*Homey et al., 2002*; *Briukhovetska et al., 2021*) was screened in TNBC tissue (12 patients- 6 long telomere, 6-short telomere). We found that *IL1R1, IL1B, TNF,* and *IL3* were lower in long telomere samples (*Figure 7F*). Patient-derived organoids (TNBC-organoids from short [TNBO-ST] or long [TNBO-LT] telomeres; *Figure 7—figure supplement 1E*) were made and tested for telomere length. We found that the expected difference in telomere length was observable between the TNBO-ST and TNBO-LT samples (*Figure 7G*). When the same set of genes as in *Figure 7F* was tested in the TNBO samples, the same trend was observed for *IL1R1, IL1B,* and *TNF*. In this case, *IL10* was also lower in TNBO-LT samples (*Figure 7H*). Secreted IL1B into the organoid media was also found to be lower in TNBO-LT samples (*Figure 7I*). For comparison, the same genes were looked at in HT1080 tumour xenograft and ex vivo cells with long/short telomeres (*Figure 7J*).

*IL1B* and *IL1R1* expression was significantly low in the case of tumours/cells with relatively long telomeres and notably consistent across clinical tissue, organoids, xenografts and cells, in contrast to the other cytokines which showed model-specific variation (*Figure 7K*). Additionally, mRNA expression correlation from 34 TNBC samples further confirmed the inverse (negative) correlation of *IL1R1* with RTL (relative telomere length) and the consistent positive correlation between *IL1R1* and *IL1B* (*Figure 7—figure supplement 1F*); showed key IL1 signalling genes largely upregulated in the short-telomere cells. Interestingly, *TRF2* showed a strong positive correlation with *IL1R1* and *IL8* but not with other pro-inflammatory cytokines.

We, next, used publicly available TCGA gene expression data of breast cancer samples (BRCA; *Supplementary file 4*) to assess the effect of *IL1R1* expression on cancer prognosis. We categorised samples based on *IL1R1* expression: *IL1R1* high (N=254) and *IL1R1* low samples (N=709). Notably, overall patient survival was significantly lower in *IL1R1* high samples (Log-rank p value: 0.0149; *Figure 7—figure supplement 1G*). We also checked the frequency of occurrence of various breast cancer sub-types in *IL1R1* high and low samples (*Figure 7—figure supplement 1H*). While invasive mixed mucinous carcinoma (the most abundant sub-type) was predominantly seen in *IL1R1* low samples, metaplastic breast cancer was only found within the *IL1R1* high samples. Interestingly, metaplastic breast cancer has been frequently found to be 'triple negative', that is ER-, PR-, and HER2- (*Reddy et al., 2020*).

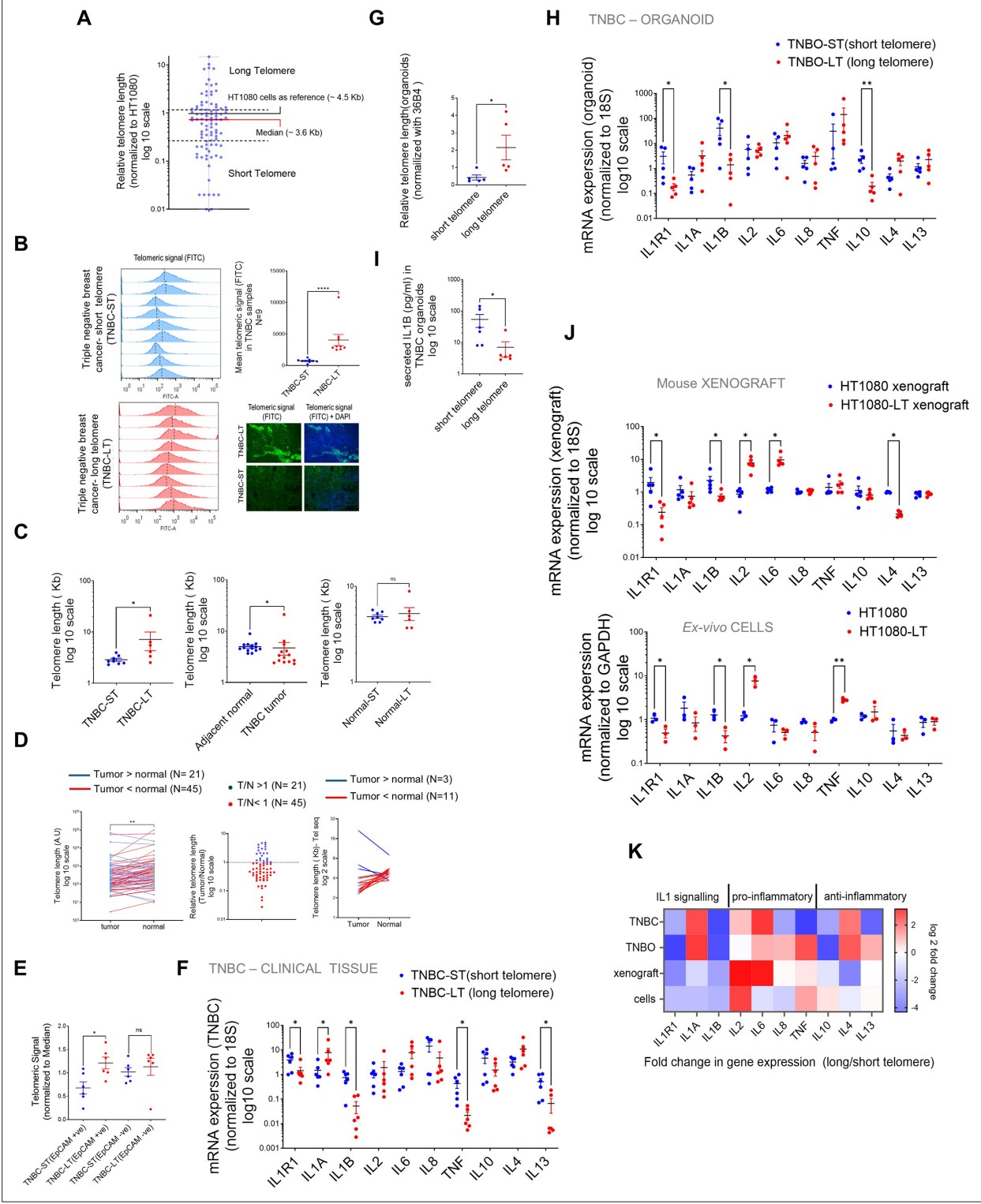

**Figure 7.** IL1R1 expression in Triple-Negative Breast cancer (TNBC) samples is sensitive to inter-tumoral variation in telomere length. (**A**) Relative telomere length of TNBC samples (94 patients) by Telo-qPCR as reported earlier; signal from telomere-specific primers was normalised to single copy number gene 36B4 for individual samples. All samples were run with HT1080 DNA as control and telomeric signal from HT1080 cells (telomere length ~4.5 Kb) was used as reference for relative measurement; median telomere length (3.6 Kb) shown by red bar; samples >50% or<50% of median (dotted lines) were designated as long or short telomere samples, respectively. [N=94]. (**B**) Flow cytometry analysis of telomere length of TNBC tissues from telomeric signal using telomere–specific FITC-labelled PNA probe; quantification of mean telomeric signal for nine TNBC-ST (short telomere, top left panel) and nine TNBC-LT (long telomere bottom left panel) shown in top right panel. Median telomere length has been indicated by dotted

*Figure 7 continued on next page*

*Figure 7 continued*

lines (top right). TNBC tissue slides hybridised with telomere-specific PNA probes and counter stained with DAPI. Representative images for long/short telomere TNBC tissue shown (bottom right) [N=9] Statistical significance was calculated using Mann-Whitney's non-parametric test (p values: *≤0.05, **≤0.01, ***≤0.001, ****≤0.0001). (C) Telomere length was determined using the previously published algorithm Tel-Seq from sequenced genomes of TNBC samples (N=8 for TNBC-ST (short telomere) and N=6 for TNBC-LT (long telomere); and adjacent normal tissue from same patient). Samples identified as long-telomere (TNBC-LT) or short-telomere (TNBC-ST) using telo-qPCR were significantly different in TL, and consistent with telo-qPCR annotation (left); TL of tumour samples was lower that adjacent normal tissue [N=15] (center); and TL from adjacent normal tissues in LT or ST samples did not vary significantly [N=8; N=6](right) Statistical significance was calculated using Mann-Whitney's non-parametric test (p values: *≤0.05, **≤0.01, ***≤0.001, ****≤0.0001). (D) Paired telomere length assessment in tumour and adjacent normal tissue for 66 patients. Telomere signal was normalised to 36B4 (single copy gene) by qRT PCR and compared between tumour and adjacent normal samples. The data was plotted pairwise (left) with lines connecting the individual tumor sample to the respective adjacent normal. All samples where tumor telomere length was lower than normal were connected by red lines and the samples where tumour telomeres where higher were connected by blue lines. Relative telomere length (Tumour/Normal) was plotted (middle) with samples with T/N<1 in red and T/N>1 in blue. Out of the 66 samples, 45 samples had T/N<1, suggesting that about two-third of the samples tested had shorter telomeres in the tumours. Analysis of whole genome sequences of 14 pairs of samples using tel-seq pipeline revealed that 11 cases showed lower tumour telomere length compared to their adjacent normal counterparts (right). Left panel-Statistical significance was computed by Wilcoxon matched-pairs signed rank test (p values: *≤0.05, **≤0.01, ***≤0.001, ****≤0.0001). (E) TNBC tissue (N=6 each for long or short telomere samples) stained with EpCAM (far red) and hybridised with telomere-specific PNA probe (FITC). Mean telomeric signal was plotted in total EpCAM$^{+ve}$ or EpCAM$^{-ve}$ cells. [N=6] Statistical significance was calculated using Mann-Whitney's non-parametric test (p values: *≤0.05, **≤0.01, ***≤0.001, ****≤0.0001). (F) mRNA expression of *IL1R1* and other key cytokines: *IL1A, IL1B, IL2, IL6, IL8, TNF, IL10, IL4,* and *IL13* in long or short telomere TNBC tissue; the *18* S gene was used for normalization. [N=6] Statistical significance was calculated using Mann-Whitney's non-parametric test (p values: *≤0.05, **≤0.01, ***≤0.001, ****≤0.0001). (G) Relative telomere length of triple-negative breast tumour organoids (TNBO) derived from TNBC samples: five TNBO-ST (short telomere) and five TNBO-LT (long telomere) was estimated by telo-qRTPCR. [N=5] Statistical significance was calculated using Mann-Whitney's non-parametric test (p values: *≤0.05, **≤0.01, ***≤0.001, ****≤0.0001). (H) mRNA expression of *IL1R1* and other key cytokines in triple-negative breast tumour organoids (TNBO) derived from TNBC samples- TNBO-ST and TNBO-LT; the *18*S gene was used for normalisation. [N=5] Statistical significance was calculated using Mann-Whitney's non-parametric test (p values: *≤0.05, **≤0.01, ***≤0.001, ****≤0.0001). (I) Secreted IL1B (pg/ml) from organoids TNBO-ST or TNBO-LT by ELISA using media supernatant form organoid culture (see Materials and methods).[N=6] Statistical significance was calculated using Mann-Whitney's non-parametric test (p values: *≤0.05, **≤0.01, ***≤0.001, ****≤0.0001). (J) mRNA expression of *IL1R1* and other key cytokines using xenograft tumours made from either short (HT1080) or long telomere (HT1080-LT) fibrosarcoma cells, and ex vivo HT1080 or HT1080-LT cells. The *18*S gene was used for normalisation for the tumours;for ex vivo cells *GAPDH* was used for normalisation. Top panel- [N=5] Statistical significance was calculated using Mann-Whitney's non-parametric test (p values: *≤0.05, **≤0.01, ***≤0.001, ****≤0.0001).; bottom panel- [N=3] Statistical significance was calculated using unpaired T test with Welch's correction (p values: *≤0.05, **≤0.01, ***≤0.001, ****≤0.0001). (K) Heat map summarizing the mRNA expression of key cytokines across the models shown above: TNBC tissue, TNBC derived organoids (TNBO), xenograft tumours or ex vivo cells in long or short telomere cases. Fold change of gene expression (long with respect to short telomeres) color coded as per the reference legend along y-axis. Error bars correspond to SEM from independent experiments.

The online version of this article includes the following source data and figure supplement(s) for figure 7:

**Source data 1.** Source data for all plots in *Figure 7* and corresponding details of statistical tests and p-values.

**Figure supplement 1.** IL1R1 expression in Triple-Negative Breast cancer (TNBC) samples is sensitive to inter-tumoral variation in telomere length (continued).

**Figure supplement 1—source data 1.** Source data for all plots in *Figure 7—figure supplement 1* and corresponding details of statistical tests and p-values.

## Telomere length sensitive IL1R1 expression modulates TAM infiltration in TNBC

Previous studies have implicated many telomere-sensitive genes to be related to immune regulation (*Barthel et al., 2017*; *Hirashima et al., 2013*). A recent study analyzed transcriptomic data across 31 cancer types to identify genes that are altered with telomere length (*Barthel et al., 2017*). The authors found that while sub-telomeric genes (up to 10 MB from the nearest telomeres) were altered with changes in telomere length as previously ascribed to the Telomere position effect (TPE; *Baur et al., 2001*; *Kim et al., 2016*; *Pedram et al., 2006*; *Robin et al., 2015*), many 'telomere-sensitive' genes were distributed far from telomeres throughout the genome. This is consistent with a previous report that telomeres can affect transcription in regions far beyond sub-telomeres independent of TPE (*Mukherjee et al., 2018*). An intriguing observation made by this study was that when the top 500 genes to be altered with telomere elongation were functionally segregated, the highest number of genes fell under the category of 'Immune response'. When a Gene Ontology analysis was done by us using these genes, 'macrophage inhibitory factor signalling' was the top biological process (*Figure 8—figure supplement 1A*). Recent studies have implicated IL1R1-related signalling in M2 TAM infiltration in tumours (*Chen et al., 2023*; *Zhang et al., 2022*). In our data, we had observed

consistent change in *IL1R1* and *IL1B* in long versus short telomere tumours and also noted that IL1 signalling has been widely implicated with TAM infiltration in multiple cancers (*Carmi et al., 2009*; *Chittezhath et al., 2014*; *Lappano et al., 2020*; *Mantovani et al., 2019*).

When breast cancer transcriptomic data from TCGA was segregated into high and low *IL1R1* groups (*Supplementary file 4*). Within the differentially expressed genes between high and low *IL1R1* groups, we looked at key immune markers such as reported TAM markers (*CD163, MRC1,D86,CD80*), TAM chemo-attractants (*CCL2, CCL3,CCL5, CCL8,CXCL12*; *Unver, 2019*) as well as reported TIL markers (*CD3G, CD4,CD8A, FOXP3,ENTPD1, PDCD1*; *Figure 8A*, *Supplementary file 5*). We found that *MRC1* (gene coding for CD206-M2 macrophage marker) was the most significantly upregulated marker in *IL1R1* high samples. Interestingly, *TERT* (often correlated/associated with high telomere length; *Greider, 1998*; *McNally et al., 2019*) was found to be significantly downregulated in *IL1R1* high samples. This observation aligns with our finding that *IL1R1* was higher in short telomere samples in TNBC and TNBO.

With these in mind, we checked TAM infiltration in TNBC-ST vis-à-vis TNBC-LT. Nine TNBC-ST and nine TNBC-LT tumours were analyzed for M2-type TAMs by double-positive CD11b (monocyte-derived cells; *Cassetta et al., 2016*) and CD206 (M2-specific; *Jaynes et al., 2020*) labelling: TNBC-LT had relatively low TAM infiltration (mean 1.78%, range: 0.18–5.41%) than TNBC-ST (mean 7.34%, range: 3.76–16.80%; *Figure 8B*; *Figure 8—figure supplement 1B*). Accordingly, CD206 expression was found to be lower in tissue sections from TNBC-LT relative to TNBC-ST tumours (*Figure 8C*). We further found that the total proportion of immune cells (% of CD45 +ve) did not vary significantly between short and long telomere TNBC samples (*Figure 8—figure supplement 1C*). However, TNBC-ST samples had a higher percentage of myeloid cells (CD11B+ve) within the CD 45+ve immune cell population. We checked in three TNBC-ST and TNBC-LT samples and found that the percentage of M1 macrophages (CD86 high CD 206 low) in the myeloid population was lower than that of the M2 macrophages (CD 206 high CD 86 low) and unlike the latter, did not vary significantly between TNBC-ST and TNBC-LT samples (*Figure 8—figure supplement 1C*).

We used the available data for *TERT/TERC* expression for a subset of the samples and plotted the telomerase activity for these TNBC samples (*Figure 8—figure supplement 1D*). Upon checking we found that while *TERT* did not significantly correlate with either telomere length or telomerase activity, the correlation with *TERC* was significant with telomere length. *TERC* was earlier shown to vary significantly between short and long-telomere TNBC samples (*Figure 7—figure supplement 1D*). Therefore, in TNBC, *TERC* seems to be critical for telomere length. Individually, neither *TERT* nor *TERC* correlated significantly with % TAM in TNBC tissue. As expected from our other results, % TAM correlated significantly with RTL. This indicates that in TNBC tissue, telomere length is possibly a more important determinant for TAM infiltration than either *TERT*, *TERC* or telomerase activity.

TAM infiltration was further tested using patient tumour-derived organoids from five long and five short telomere cases. M2-type macrophage derived from THP1 cells (*Figure 8—figure supplement 1E*) were labelled (red-tracker dye) before co-incubating with each of the five TNBO-ST or five TNBO-LT: Following 12 hr incubation TNBO-ST had on average 8.56% infiltration of M2-type macrophages (range: 6.54–10.9%) compared to 3.14% (range: 0.99–4.88%) for TNBO-LT (*Figure 8D*).

Further, we used the IL1R1-receptor antagonist IL1RA to directly test IL1R1 dependence. The minimum dosage of IL1RA where expression of inflammatory markers was found to be altered in TNBO was used in the assays (*Figure 8—figure supplement 1F*). Organoids treated with IL1RA before co-incubating with dye-labelled M2-type macrophages showed >50% reduction in macrophage infiltration (*Figure 8E*).

Finally, we screened several G4 binding ligands in MDAMB231 cells and found that a few ligands (JD83, 12459, and NMM) could reduce *IL1R1* expression (*Figure 8—figure supplement 1G*). These ligands were tested in TNBO and we found JD83 to be effective in reducing *IL1R1* expression at 5 µM concentration (*Figure 8F*). We found that JD83 is also effective in reducing M2 migration in TNBO, showing promise for future therapeutic interventions.

## Discussion

Recently, we described the TL-sensitive expression of genes distant from telomeres (*Mukherjee et al., 2018*). Cells with relatively long telomeres sequestered more TRF2 at the telomeres affecting non-telomeric binding, and vice-versa. Proposed as the telomere-sequestration-partition (TSP)

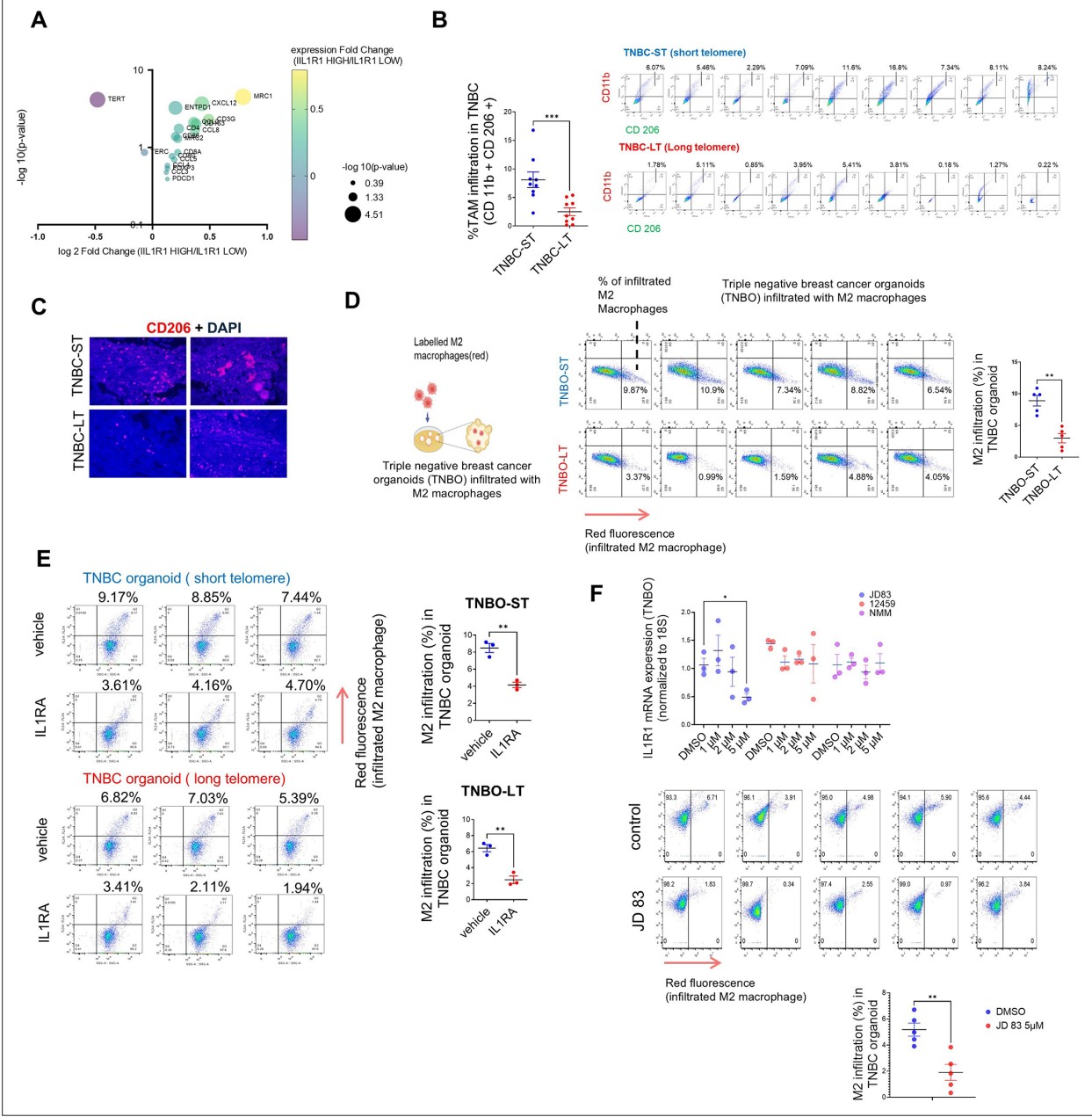

**Figure 8.** Telomere length sensitive IL1R1 expression modulates tumour associated macrophage (TAM) infiltration in TNBC. (**A**) Key Immune marker expression (TAM markers, TAM related chemokines, T cell markers), *TERT* and *TERC* were plotted for fold change of transcript level expression (IL1R1 high / IL1R1 low) and adjusted p-values in breast cancer (BRCA) samples categorized as IL1R1 high or low from TCGA (see ***Supplementary file 4***). (**B**) Percentage TAM infiltration in TNBC tissue with short (TNBC-ST) or long (TNBC-LT) telomeres using markers CD11b and CD206 (CD11b+CD206+ cells shown in top right quadrant); quantification of TAM infiltration from individual TNBC-ST or TNBC-LT samples have been plotted. [N=9] Statistical significance was calculated using Mann-Whitney's non-parametric test (p values: *≤0.05, **≤0.01, ***≤0.001, ****≤0.0001). (**C**) TNBC tissue slides stained with TAM-specific marker CD206 for macrophage infiltration within tissue (counterstained with DAPI); representative images for two independent long or short telomere TNBC tissues shown. (**D**) Labelled (red) M2 macrophages derived from THP1 cells incubated with triple-negative breast tissue-derived organoids (TNBO) for 12 hr (scheme in left panel). Infiltration of M2 macrophages in organoids shown as percentage of labelled (red) M2 macrophages in individual flow-cytometry plots (florescence signal (FL3) in x-axis in log scale); percentage infiltration from five TNBO-ST or five TNBO-LT samples shown in right panel. [N=5] Statistical significance was calculated using Mann-Whitney's non-parametric test (p values: *≤0.05, **≤0.01, ***≤0.001, ****≤0.0001). (**E**) Labelled (red) M2 macrophage infiltration in presence/absence of receptor antagonist IL1RA (20 ng/ml) for tumour organoids with short (TNBO-ST) or long telomeres (TNBO-LT) in three replicates; percentage values for M2 infiltration plotted in right panel. Red florescence (FL3; y-axis in log scale) and respective percentage M2 infiltration values marked on top of individual flow cytometry plots. [N=3] Statistical significance was calculated using unpaired T test with Welch's correction (p values: *≤0.05, **≤0.01, ***≤0.001, ****≤0.0001). (**F**) mRNA expression of *IL1R1* was checked

*Figure 8 continued on next page*

*Figure 8 continued*

in presence of varying concentrations of G-quadruplex binding ligands (48 hr treatment) using TNBO. *18* S gene was used for normalisation. Following this, the ligand JD83 was used to check M2 infiltration in TNBO as in (**C, D**). The M2 infiltration has been plotted for control and JD 83 treated samples. Top panel- [N=3] Statistical significance was calculated using unpaired T test with Welch's correction (p values: *≤0.05, **≤0.01, ***≤0.001, ****≤0.0001).; bottom panel- [N=5] Statistical significance was calculated using Mann-Whitney's non-parametric test (p values: *≤0.05, **≤0.01, ***≤0.001, ****≤0.0001). Error bars correspond to SEM from independent experiments.

The online version of this article includes the following source data and figure supplement(s) for figure 8:

**Source data 1.** Source data for all plots in *Figure 8* and corresponding details of statistical tests and p-values.

**Figure supplement 1.** Telomere length sensitive IL1R1 expression modulates tumour associated macrophage (TAM) infiltration in TNBC (continued).

**Figure supplement 1—source data 1.** Source data for all plots in *Figure 8—figure supplement 1* and corresponding details of statistical tests and p-values.

model (*Vinayagamurthy et al., 2020*), this is independent of telomere-positioning-effect (TPE) on expression of sub-telomeric genes (up to 10 Mb from telomeres; *Robin et al., 2014*). TL-dependent *IL1R1* activation by non-telomeric TRF2 is in tune with the TSP model. It needs to be noted that the *IL1R1* gene is located above 100 MB away from the nearest telomere in humans, and hence far out of the proposed 10 MB range of the TPE model. Altered transcription when telomere maintenance is affected in varied contexts (*Hirashima et al., 2013*; *Mukherjee et al., 2018*), particularly across 31 cancer types (*Barthel et al., 2017*), appears consistent with TSP and support our observations

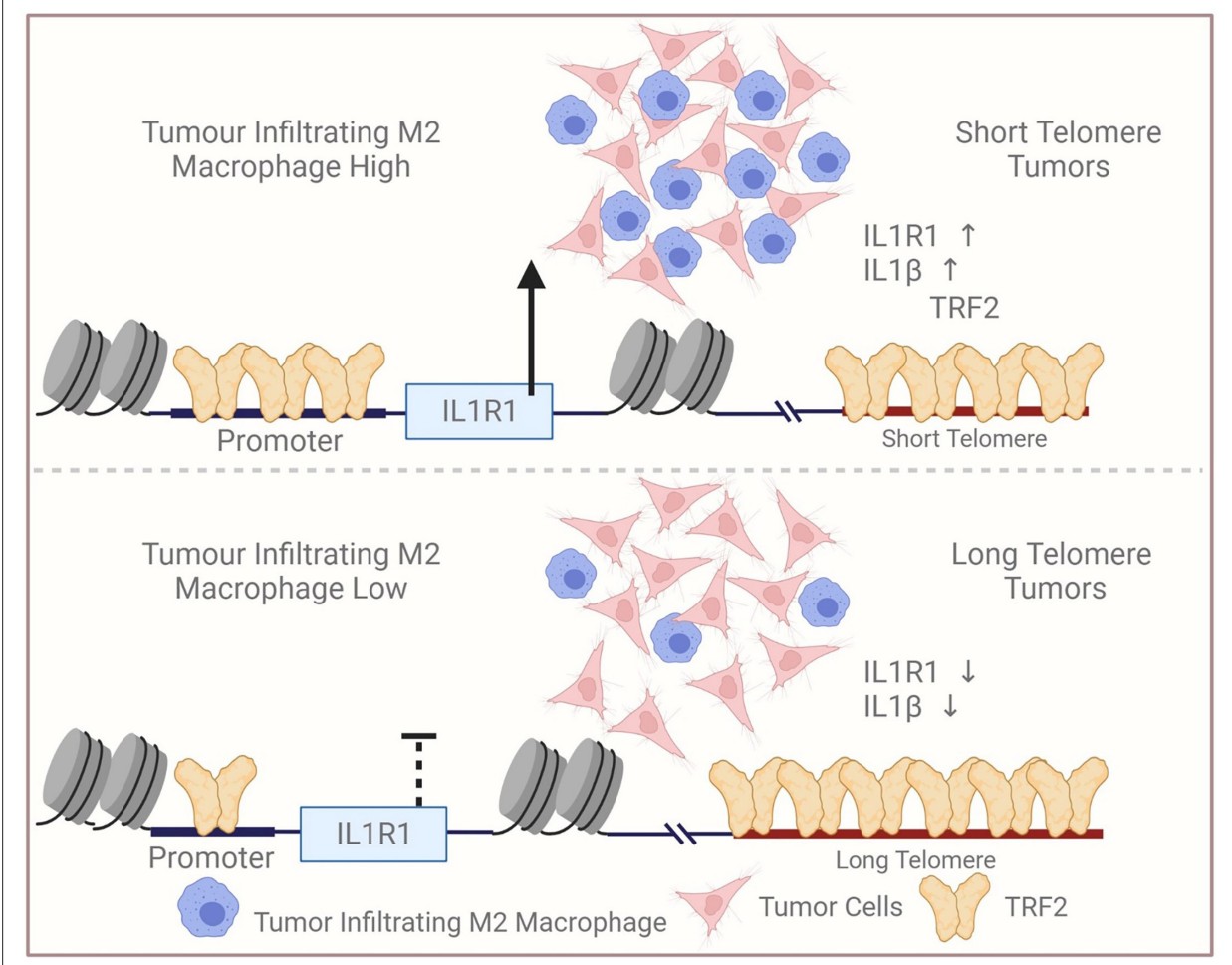

**Figure 9.** Graphical summary. Scheme depicting relatively low infiltration of TAM in tumours with relatively long telomeres vis-à-vis tumours with short telomeres. Reduced non-telomeric TRF2 binding at the IL1R1 promoter in tumours with long telomeres, and consequent low IL1R1 activation, attenuated p65-mediated IL1-beta and macrophage infiltration. Created with BioRender.com.

related to TL-dependent IL1 signalling. A model of telomere length and TRF2-dependent *IL1R1* expression and the potential modulation of TAM infiltration is presented as a graphical summary (*Figure 9*).

Besides TRF2, the non-telomeric function of the shelterin protein RAP1 has been reported as part of complex with NF-kappaB (*Teo et al., 2010*). Although TRF2 has been implicated in influencing Natural Killer (NK) cells and infiltration of myeloid derived immuno-suppressive cells (MDSC; *Biroccio et al., 2013*; *Cherfils-Vicini et al., 2019*) underlying mechanisms were unclear. However, the function of non-telomeric TRF2 in either IL1 signalling or in a telomere-sensitive role in immune signalling, has not been previously studied.

Mechanistically, we found that DNA secondary structure G4s present on the *IL1R1* promoter are necessary for TRF2 binding. Perturbation of the G4s genetically by destabilizing base substitutions, or upon using small molecules that bind to G4 inside cells resulted in reduced promoter TRF2 binding and *IL1R1* expression. This is in line with our earlier results showing G4-dependent non-telomeric TRF2 binding throughout the genome, and how TRF2 recruitment to specific promoters leads to epigenetic histone modifications (*Mukherjee et al., 2019a*; *Mukherjee et al., 2018*; *Sharma et al., 2021*). We recently noted the transcription repression of telomerase (*hTERT*) to be TRF2-mediated in a PRC2-repressor-dependent fashion (*Sharma et al., 2021*). This, interestingly, suggests the emerging role of TRF2 as a transcriptional modulator with repressor/activator (as shown here) functions depends on cofactor engagement that is possibly contextual.

The histone-modifying enzyme complex-p300/CBP have been keenly studied in cancer epigenetics specifically breast cancers (*Iyer et al., 2004*; *Ramadan et al., 2021*; *Ring et al., 2020*). Recent findings also show acetylated H3K27 enrichment at the telomeres through p300-dependent HAT activity (*Cubiles et al., 2018*); however, role of TRF2 in p300 engagement or H3K27 acetylation at the telomeres was not investigated. Here, notably, we found TRF2-dependent p300 recruitment, and HAT activity resulting in H3K27 acetylation at the *IL1R1* promoter. Together, these support the role of non-telomeric TRF2 as an activating transcription factor through p300 implicating a broader role of TRF2-p300-dependent histone modifications in cancer. Further, these suggest the possibility of similar functions of telomeric TRF2 in H3K27 acetylation at the telomeres (*Cubiles et al., 2018*). The lysine residue K293 acetylation on TRF2 seems to be crucial for p300-TRF2 interaction.

It has been noted previously that TERT can have effects on NF-kappa B signalling independent of telomere length alterations (*Ghosh et al., 2012*). We observed that in long telomere HT1080-LT cells, while NF-kappa B signalling was retarded- a late surge of NFKappa-B activation occurred. We are inclined to think this could be a potential effect of TERT over-expression. Understanding the extent to which TERT influences the inflammatory signalling in TNBC independent of telomere length would require further work.

IL1 signalling in relation to inflammation in cancer has been extensively studied and consequently, IL1B blockers and the receptor antagonist IL1RA are of clinical interest (*Homey et al., 2002*; *Chittezhath et al., 2014*; *Balkwill and Mantovani, 2001*; *Kaplanov et al., 2019*; *Lappano et al., 2020*). Further, the effect of altered IL1 signalling on M2-type TAM infiltration was reported in multiple cancer types (*Chanmee et al., 2014*; *Hu et al., 2016*). TAM infiltration has been consistently linked to tumour aggressiveness (*Oner et al., 2020*) and recent work on targeting (*Lee et al., 2019*; *Rogers and Holen, 2011*) and modifying TAMs from M2-like states to a more anti-tumourigenic M1-like states (*Jaynes et al., 2020*) have shown promise as strategies for intervention in cancer. The clinical significance of blocking IL1 signalling in cancers has been widely reported (*Briukhovetska et al., 2021*; *Kaplanov et al., 2019*; *Mantovani et al., 2018*). Based on our findings, we attempted TRF2-mediated single-gene-targeted epigenetic silencing of *IL1R1* in cell lines and ligand-based approaches in TNBC organoids. A combination of single gene editing and small molecule/ ligand-based intervention might be useful in therapy.

For the functional significance of TL in TME, we focused on IL1-signalling and TAM. Results from TNBC organoids here show that blocking IL1R1 using IL1RA was sufficient for restricting M2 infiltration (*Figure 8E*) suggesting that the IL1 signalling within the tumour cell is important. However, particularly because macrophages are known to be a rich source of IL1B (*Carmi et al., 2009*; *Arango Duque and Descoteaux, 2014*; *Mantovani et al., 2018*), and given the model-specific changes in inflammatory cytokines noted by us, further experiments will be required to understand how telomeres influence paracrine/ juxtracrine IL1 signalling between cancer cells and TAM.

It is quite possible that TL affects tumour-immune signalling through other immune cell types as well [implied by altered immune-response genes in TL-dependent gene-expression analysis (*Barthel et al., 2017*; *Hirashima et al., 2013*)] and/or influences the nature of TAM (*Hung et al., 2016*), which is known to vary contextually (*Mantovani and Locati, 2013*). These possibilities, together with the well-established effect of IL1-targeting in improving cancer prognosis (*Briukhovetska et al., 2021*; *Cristofari and Lingner, 2006*; *Mantovani et al., 2019*; *Mantovani et al., 2018*) underline the need for further work to attain a deeper understanding of telomere-sensitive IL1 signalling in the TME.

Although we observed a range of telomere lengths across cancers in TNBC patients (*Figure 7A–C*), as also noted by other groups (*Londoño-Vallejo, 2004*; *Pellatt et al., 2013*; *Shen et al., 2007*; *Wentzensen et al., 2011*; *Willeit et al., 2010*), tumours largely have been reported to maintain shorter telomeres than normal tissue (*Aviv et al., 2017*). Here, we found that tumour cells with shorter TL-activated IL1 signalling increase M2-type TAM infiltration, which is known to aid immunosuppression (*Allavena et al., 2008*). Based on our results, and other reports suggesting immunomodulation in cancers with relatively short telomeres (*Barthel et al., 2017*; *Hirashima et al., 2013*), it is therefore tempting to speculate that short-telomere cancer cells might be 'selectively' better equipped to evade host immune cells. To sum up- our work reveals new molecular connections between telomeres and tumour immunosuppression, presenting a conceptual framework for a better understanding of patient-specific responses to cancer immunotherapy.

# Materials and methods

**Key resources table**

| Reagent type (species) or resource | Designation | Source or reference | Identifiers | Additional information |
|---|---|---|---|---|
| Cell line (*Homo sapiens*) | HT0180 | Human | ATCC -CCL-121 | Fibrosarcoma |
| Cell line (*H. sapiens*) | MDAMB231 | Human | ATCC HTB-26 | Breast cancer |
| Cell line (*H. sapiens*) | HEK293T | Human | ATCC CRL-3216 | Embryonic kidney derived |
| Cell line (*H. sapiens*) | MRC5 | Human | ATCC CCL-171 | Immortalized fibroblast |
| Cell line (*H. sapiens*) | THP1 | Human | ATCC TIB-202 | Monocyte derived from peripheral blood from an acute monocytic leukemia patient |
| Peptide, recombinant protein | R-Spondin 3 | R&D | 3500-RS/CF | TNBC organoid culture reagent |
| Peptide, recombinant protein | Neuregulin 1 | Peprotech | 100–03 | TNBC organoid culture reagent |
| Peptide, recombinant protein | FGF 7 | Peprotech | 100–19 | TNBC organoid culture reagent |
| Peptide, recombinant protein | FGF 10 | Peprotech | 100–26 | TNBC organoid culture reagent |
| Peptide, recombinant protein | EGF | Peprotech | AF-100–15 | TNBC organoid culture reagent |
| Peptide, recombinant protein | Noggin | Peprotech | 120–10 C | TNBC organoid culture reagent |
| Other | A83-01 | Tocris | 2939 | TNBC organoid culture reagent |
| Other | Y-27632 | Abmole | Y-27632 | TNBC organoid culture reagent |
| Other | SB202190 | Sigma | S7067 | TNBC organoid culture reagent |
| Other | B27 supplement | Gibco | 17504–44 | TNBC organoid culture reagent |
| Other | N-Acetylcysteine | Sigma | A9165-5g | TNBC organoid culture reagent |
| Other | Nicotinamide | Sigma | N0636 | TNBC organoid culture reagent |
| Other | GlutaMax 100 x | Invitrogen | 12634–034 | TNBC organoid culture reagent |

*Continued on next page*

*Continued*

| Reagent type (species) or resource | Designation | Source or reference | Identifiers | Additional information |
|---|---|---|---|---|
| Other | Hepes | Invitrogen | 15630–056 | TNBC organoid culture reagent |
| Other | Penicillin/Streptomycin | Invitrogen | 15140–122 | TNBC organoid culture reagent |
| Other | Primocin | Invivogen | Ant-pm-1 | TNBC organoid culture reagent |
| Other | Advanced DMEM/F12 | Invitrogen | 12634–034 | TNBC organoid culture reagent |
| Antibody | TRF2 (rabbit polyclonal) | Novus | (Novus NB110-57130)- | ChIP (1:100), IP (1:100), WB (1:1000) |
| Antibody | TRF2 (mouse monoclonal) | Millipore | (Millipore 4A794)- | Flow cytometry (1:500) |
| Antibody | TRF1 (mouse monoclonal) | Novus | (Novus NB110-68281)- | WB (1:1000), IP (1:100) |
| Antibody | IL1R1 (rabbit polyclonal) | Abcam | (abcam ab106278)- | IF (1:500), Flow cytometry (1:500), WB (1:1000) |
| Antibody | DDK/FLAG (mouse monoclonal) | Sigma | (Sigma-F1804)- | ChIP (1:100), WB (1:1000) |
| Antibody | BG4 (recombinant antibody) | Sigma | (Sigma Aldrich MABE917)- | ChIP (1:100) |
| Antibody | P300 (rabbit monoclonal) | CST | (CST D2X6 N) | ChIP (1:100), IP (1:100), IF (1:1000), WB (1:1000) |
| Antibody | Ac-p300/CBP (rabbit monoclonal) | CST | (CST 4771) | ChIP (1:100), IP (1:100), WB (1:1000) |
| Antibody | CBP (rabbit monoclonal) | CST | (CST 7389) | ChIP (1:100), WB (1:1000) |
| Antibody | P65(NFKB) (rabbit monoclonal) | CST | (CST 8242) | WB (1:1000) |
| Antibody | Ser536 P p65 NFKB (rabbit monoclonal) | CST | (CST 3033) | WB (1:1000) |
| Antibody | IKB (mouse monoclonal) | CST | (CST 4814) | WB (1:1000) |
| Antibody | p-IKB (rabbit monoclonal) | CST | (CST 2589) | WB (1:1000) |
| Antibody | CD44 (mouse monoclonal) | CST | (CST 3570) | IF (1:1000) |
| Antibody | Histone H3 (rabbit polyclonal) | Abcam | (Abcam ab1791)- | ChIP (1:100) |
| Antibody | H3K27ac (rabbit polyclonal) | Abcam | (Abcam ab4729)- | ChIP (1:100) |
| Antibody | H3K4me3 (rabbit polyclonal) | Abcam | (Abcam ab8580)- | ChIP (1:100) |
| Antibody | H3K27me3 (rabbit monoclonal) | Abcam | (abcam ab192985)- | ChIP (1:100) |
| Antibody | H3K9me3 (rabbit polyclonal) | Abcam | (abcam ab8898)- | ChIP (1:100) |
| Antibody | Beta-actin (mouse monoclonal) | Santa-cruz | (Santacruz C4)- | WB (1:1000) |
| Antibody | GAPDH (mouse monoclonal) | Santa-cruz | (Santacruz 6C5)- | WB (1:1000) |

*Continued on next page*

*Continued*

| Reagent type (species) or resource | Designation | Source or reference | Identifiers | Additional information |
|---|---|---|---|---|
| Antibody | CD11b (mouse monoclonal) | eBioscience | (ICRF44), eBioscience- | Flow cytometry (1:500) |
| Antibody | CD206 (mouse monoclonal) | Invitrogen | (MR5D3) Invitrogen- | Flow cytometry (1:500) |
| Antibody | APC anti-human CD326 (EpCAM) (mouse monoclonal) | Biolegend | (9C4) BioLegend- | Flow cytometry (1:500) |
| Antibody | FITC anti-human CD11B (mouse monoclonal) | Biolegend | (ICRF44) BioLegend- | Flow cytometry (1:500) |
| Antibody | PE anti-human CD45 (mouse monoclonal) | Biolegend | (HI30) BioLegend- | Flow cytometry (1:500) |
| Antibody | PE-Cy7 anti-human CD45 (mouse monoclonal) | Biolegend | (W17233E) BioLegend- | Flow cytometry (1:500) |
| Antibody | Anti-Rabbit IgG (rabbit polyclonal) | Millipore | (Millipore 12-370) | isotype control (1:100) |
| Antibody | Anti-mouse IgG (mouse polyclonal) | Millipore | (Millipore 12–371) | isotype control (1:100) |
| Recombinant DNA reagent | pCMV6-myc-DDK(FLAG)-TRF2 vector (TRF2 WT) | origene | RC223601 | Plasmid |
| Recombinant DNA reagent | TRF2 shRNA | origene | TL308880 | Plasmid |
| Recombinant DNA reagent | dCAS9-VP64-GFP | Addgene | 61422 | Plasmid |
| Recombinant DNA reagent | pX333 | Addgene | 64073 | Plasmid |
| Recombinant DNA reagent | pENTR11 | Invitrogen | K253520 | Plasmid |
| Recombinant DNA reagent | pCW57.1 | Addgene | 41393 | Plasmid |
| Recombinant DNA reagent | TERC shRNA | Santa-cruz | sc-106994-SH | Plasmid |
| Recombinant DNA reagent | PGL3 basic | Promega | E1751 | Plasmid |
| Recombinant DNA reagent | pGL4.73 | Promega | E611 | Plasmid |

## Cell lines, media, and culture conditions

HT1080 fibrosarcoma derived cell line was purchased from ATCC. HT1080-LT cells (*Cristofari and Lingner, 2006*) were a kind gift from Dr. J. Ligner. HT1080 and MRC5 (purchased from ATCC) cells and corresponding telomere elongated cells were maintained in Modified Eagle's medium (MEM; Sigma-Aldrich) supplemented with 10% Fetal Bovine Serum (FBS; Gibco). MDAMB231 (gift from Mayo Clinic, Minnesota, USA) and HEK293T (purchased from NCCS, Pune) cells were maintained in DMEM-HG (Dulbecco's Modified Eagle's medium - High Glucose; Sigma-Aldrich) with a supplement of 10% FBS. All THP1 cell lines (gift from Dr Vivek Rao, IGIB, Delhi and subsequently purchased from NCCS, Pune) and derivative cell types were maintained in RPMI with 10% FBS and 1 X Anti-Anti (anti-biotic and anti-mycotic from Thermo Fisher Scientific). All cultures were grown in incubators maintained at 37 °C with 5% $CO_2$. All stable lines generated from these cell types were maintained in the same media with 1 X Anti-Anti.

## Processing of tumour/xenograft tissue

Tissues collected for this study were stored in RNA later (Sigma-Aldrich) at –80 °C. Approximately 10 milligrams of tumour tissue were weighed and washed thrice with ice-cold 1 X PBS- 1 ml (filtered).

About 300 μl of PBS was left to submerge the tissue after the final wash. Using a sanitised razor blade, the tumour was cut into small pieces (the efficiency of the digestion is dependent on how well the tumour is minced; large pieces will not digest properly and may need extra digestion time). The tumour soup was transferred into a 15 ml tube using a P1000 pipette (cut the top of the tip to be able to take all the tissue) and the volume was adjusted to 4 ml of 1 X pre-chilled PBS. 80 μl of Liberase TL stock solution (Roche) was mixed using a vortex and incubated for 45 min at 37 °C under continuous rotation. The sample was checked visually; the cell suspension should look smooth (if the cell suspension still contains tissue fragments digest for an additional 15 min). To end the enzymatic digestion,10 ml PBS with 1 % w/v BSA (Sigma-Aldrich) was added. The cells were filtered using a 100 μm cell strainer in a 50 ml tube and the volume adjusted to 20 ml with 1 X PBS (pre-chilled). Cells were centrifuged for 7 min at 500 × $g$ and 4 °C. The supernatant was discarded and the cells were re-suspended in 1 ml serum-free media. Subsequently, cells were counted and volume was made up to get a cell density of 1 million cells/ml. Following this, the cell suspension was used for ChIP, DNA/RNA isolation, and flow cytometry.

## Organoid generation and culture

TNBC tissue was used to make tumour organoids using the following protocol:

1. The fresh tissue was stored in 5 ml of Media 1 (Advanced DMEM F12 +1 X Glutamax +10 mM HEPES +1 X AA).
2. 1–3 mm$^3$ tissue piece was cut, washed and minced in 10 ml (Media 1).
3. The minced tissue was digested overnight on a shaker in Media 1 with 1–2 mg/ml of collagenase.
4. After overnight incubation the digested tissue was sequentially sheared using a 1 ml pipette.
5. After each shearing step, the suspension was strained using a 100 μm filter to retain tissue pieces. Thus entering in subsequent shearing step with approximately 10 ml Advanced DMEM F12.
6. To this, 2% FCS was added and centrifuged at 400 × $g$.
7. The pellet was resuspended in 10 mg/ml cold cultures growth factor reduced BME Type 2 and 40 μl drops of BME suspension was allowed to solidify on pre-warmed 24well culture plates at 37 °C 5% $CO_2$ for 30 min.
8. Upon complete gelation 400 μl of BC organoid medium was added to each well and plates transferred to 37 °C 5% $CO_2$.

| Medium component | Supplier | Catalogue number | Final concentration |
| --- | --- | --- | --- |
| R-Spondin 3 | R&D | 3500-RS/CF | 250 ng·ml$^{-1}$ |
| Neuregulin 1 | Peprotech | 100–03 | 5 nM |
| FGF 7 | Peprotech | 100–19 | 5 ng·ml$^{-1}$ |
| FGF 10 | Peprotech | 100–26 | 20 ng·ml$^{-1}$ |
| EGF | Peprotech | AF-100–15 | 5 ng·ml$^{-1}$ [a] |
| Noggin | Peprotech | 120–10 C | 100 ng·ml$^{-1}$ |
| A83-01 | Tocris | 2939 | 500 nM |
| Y-27632 | Abmole | Y-27632 | 5 mM |
| SB202190 | Sigma | S7067 | 500 nM [b] |
| B27 supplement | Gibco | 17504–44 | 1x |
| N-Acetylcysteine | Sigma | A9165-5g | 1.25 mM |
| Nicotinamide | Sigma | N0636 | 5 mM |
| GlutaMax 100 x | Invitrogen | 12634–034 | 1x |
| Hepes | Invitrogen | 15630–056 | 10 mM |
| Penicillin/Streptomycin | Invitrogen | 15140–122 | 100 U·ml$^{-1}$/100 mg·ml$^{-1}$ |
| Primocin | Invivogen | Ant-pm-1 | 50 mg·ml$^{-1}$ |
| Advanced DMEM/F12 | Invitrogen | 12634–034 | 1x |

9. The media was changed every fourth day.
10. The culture was grown for 10 days before passaging.
II. Passaging organoid cultures

1. Cultrex growth factor reduced BME Type 2 geltrex was thawed overnight on ice at 4 °C.
2. Media was removed from the organoid wells and washed twice with PBS-EDTA.
3. For each well of 24-well plate 300 µl of TrypLE was added and incubated at 37 °C 5% $CO_2$ for 5 min.
4. 600 µl of washing media (DMEM F12) was added and organoids were dissociated vigorously by pipetting.
5. Organoid solution was transferred to 15 ml falcon with 5 ml washing media and centrifuged at 1000 rpm for 5 min to pellet organoids.
6. Supernatant was carefully aspirated leaving behind 200 µl media and 500 µl of fresh media, mixed well gently and transferred to 1.5 ml tube for centrifugation at 650 × $g$ for 5 min at 4 °C.
7. Supernatant was gently aspirated and resuspended in cold cultrex and 40 µl drop/well was seed in a pre-warmed 24well plate.
8. Incubated for 30 min and add 400 µl of BC organoid medium.

## M2 macrophage generation from THP1 cells

THP1 cells (0.5 million cells in 1 ml) were seeded in a 35 mm dish and treated with PMA (10 ng/ml; Sigma-Aldrich) for 24 hr in RPMI complete media (+10% FBS; Gibco). PMA was removed and cells were cultured for 48 hr in RPMI to get semi-adherent M0 cells. Following this, cells were treated with IL4 (10 ng/ml; Gibco) and IL13 (20 ng/ml; Gibco) for 24 hr in RPMI complete media. Cells were checked for expected morphological changes and trypsinized for use as M2 macrophages.

## M2 macrophage infiltration assay in TNBC tumour organoids

M2 macrophages were labelled with Cell Tracker red dye (Thermo Fisher Scientific) for 30 min and the excess dye was washed off. 10000 labelled M2 macrophages in 20 µl 1 X PBS were introduced in a 96-well with TNBC organoids in a dome with supplement enriched media (200 µl). 12 hr later, the media was washed off with three 1X PBS washes and the organoid dome was extracted with 100 µl TrypLE (Gibco) and resuspended with 100 µl chilled 1 X PBS. This cocktail was kept on ice for 5 min and centrifuged at 1000 × $g$ for 10 min. The supernatant was removed and the process was repeated.

The cells were resuspended in 500 µl chilled 1 X PBS and subjected to Flow-cytometric analysis for scoring percentage of labelled (red) M2 macrophages infiltrating the organoids.

## Xenograft tumour generation in NOD-SCID mice

The xenograft tumour generation in NOD SCID tumours was outsourced to a service provider (Vivo Bio TechLtd – Telangana, India). 2.5 million cells (HT1080, HT1080-LT or HT1080-TRF2 inducible) were subcutaneously injected in healthy male mice (4–6 weeks old) and tumour growth was allowed. In case of TRF2 inducible HT1080 cells, tumours were allowed to grow to 100 mm³ before oral dosage of doxycycline (or placebo control) was started at 2 mg/kg of body weight of mice. Mice were sacrificed to obtain tumours when tumours reached an average volume of 600 mm³(±100).

## Telomeric fluorescent in situ hybridization (FISH)

Telomere-specific PNA probe [TelC-FITC-F1009-(CCCTAA)n from PNABio] was used following the manufacturer protocol with minor modifications. Briefly, the cells grown in chamber slides up to ~70% confluency and were treated with RNase solution for 5–10 min in 1 X PBS at room temperature. For tissue section, rehydrated tissue sections on slides were used prior to RNase treatment. Slides were washed in chilled PBS twice followed by dehydration with an increasing concentration of ethanol gradient. Slides were briefly warmed at 80 °C and then 250 nM PNA was added with hybridisation buffer (20 mM Tris, pH 7.4, 60% formamide (Sigma-Aldich), 0.5% of 10% Goat Serum). Slides were heated and covered with the buffer at 85 °C for 10 min, kept at room temperature in a dark humid chamber for 4 hr to overnight to facilitate slow cooling and hybridisation. The unbound PNA was washed with wash buffer (2 X SSC, 0.1% Tween-20 (Thermo Fisher Scientific)) followed by PBS wash twice. DAPI solution (Sigma-Aldrich) was added for 5 min and then the slides were washed in PBS. ProLong Gold Antifade Mountant (Thermo Fisher Scientific) was added and the section was covered

with a cover glass. Imaging was done with a Leica SP8 confocal microscopy system. Images were analyzed using open source image analysis software Fiji (Image J, imagej.net/software/fiji/).

## DNA and RNA extraction from TNBC/xenograft tissue

Allprep DNA/RNA Universal kit (QAGEN) was used as per manufacturer's suggested protocol for extraction of DNA and RNA from single cell suspension prepared from tissues as described before.

## Telo-qPCR

For telomere length assessment by qRT PCR, telomere-specific primers and single copy number gene 36B4 for normalisation was used in PCR reactions with sample DNA using reported protocol (*O'Callaghan et al., 2008*).

The primers used are as follows:

Tel F CGGTTTGTTTGGGTTTGGGTTTGGGTTTGGGTTTGGGTT
Tel R GGCTTGCCCTTACCCTTACCCTTACCCTTACCCTTACCCT
36B4F CAGCAAGTGGGAAGGTGTAATCC
36B4R CCCATTCTATCATCAACGGGTACAA

The relative telomere length of TNBC patient tissue samples were estimated by normalizing to telomere signal of HT1080 fibrosarcoma cells (reference cell type) as follows:

Telomere signal = $2^{-(Ct\,Tel\,-\,Ct\,36B4)}$
Relative telomere length (RTL)=Telomere signal (sample) / Telomere signal (reference).

## Flow-cytometry

The procedure for Flow-cytometry was adapted from basic flow cytometry guidelines from CST and has been previously reported in *Sharma et al., 2021*. The cell suspension was fixed with 4% formaldehyde and permeabilised with 0.1% triton X (in 1XPBS) for nuclear proteins. Incubation of primary antibody (1:500 v/v in 1% BSA in 1 X PBS) for 1 µg/ml antibody concentration for 2 hr at RT or at 4°C overnight, was followed by PBS wash (three times) and fluorescently labelled secondary antibody (1:1000 v/v) for 2 hr at RT. Following this, cells were washed with 1 X PBS (three times) and re-suspended in 1XPBS for flow-cytometric analysis using BD Accuri C6 or BD FACSAria III. Equal number of events (10000 for most of the analysis) were recorded for calculating percentage infiltration.

For single channel analysis, histogram prolife (log scale) was generated in ungated sample-SSC-FSC plots. For quadrant gating for double positive cells (two channels), biex- scale plots or log scale were used. All statistics like mean fluorescence intensity and standard deviation were estimated using built-in tools for analysis in the software FlowJo.

## Relative telomeres length (RTL) by flow cytometry

The RTL value is calculated as the ratio between the telomere signal of each sample and the control cell. The Telomere PNA Kit/FITC kit from Dako (Agilent) was used to estimate Telomere length signal in samples. The process has been previously standardised and reported (*Mukherjee et al., 2018*).

## TelSeq- estimation of telomere length using sequencing data

We used TelSeq (*Ding et al., 2014*) to estimate the average TL for each tumour and normal sample in our dataset. TelSeq was run with default parameters using WGS (whole genome Sequencing) samples with coverage ≥30 X, except the read length parameter (-r) which was set to 150 to match with our data. Briefly, the TL was calculated as the total number of reads containing seven or more telomeric repeats ('TTAGGG') divided by the total number of reads with GC content 48–52%. Further, the resultant value was multiplied with a constant value to account for genome size and the number of telomeric ends. This was done separately for each read group within a sample, and then the weighted average was calculated for each sample by taking into account the number of reads per group (as weights).

The estimated average TL (represented in Kilobase-pair [Kb] units) for tumour and normal samples has been provided in *Figure 7—figure supplement 1B*. Overall, we found that the tumour samples have lower TL scores (Median: 3.20 Kb) as compared to the matched normal samples (Median:

4.67 Kb). Also, the TL estimates from our tumour samples (Median: 3.20 Kb, range:2.3–21.10 Kb) were comparable to the TCGA breast cancer cohort (WGS; Median: 3.5 Kb, range: 1.4–29.7 Kb) analyzed by *Barthel et al., 2017*.

To check if the tumour samples with high TL estimates were confounded by the aneuploidy and tumour purity (which was not taken into account in the TelSeq analysis), we computed the purity and ploidy values for each tumour sample (where possible) using ASCAT (*Raine et al., 2016*) through Sarek pipeline (*Garcia et al., 2020*). However, we found no significant difference in the purity (Wilcoxon test two-sided p=0.25) and ploidy (Wilcoxon test two-sided p=0.52) values between tumour samples with high versus low TL scores, suggesting that the tumours with high TL estimates were not influenced by these parameters.

## IL1B and IL1RA treatment

Reconstituted IL1B (Sigma-Aldrich) was treated to appropriate cells at 10 ng/ml concentration followed by assays. Similarly, IL1RA (Abcam) was treated to appropriate cell/TNBO at 20 ng/ml concentration followed by assays at appropriate time points in specific experiments.

## IL1B ELISA

IL1B ELISA was performed using Human IL-1 beta/IL-1F2 DuoSet ELISA kit by R&D Biosystems. As per manufacturer's protocol 96-well microplate was coated with100ul per well of freshly prepared Capture Antibody diluted 100 times in PBS without carrier protein. Thereafter the plate was sealed and incubated overnight at room temperature. The following day, the capture antibody was aspirated after which each well was washed three times with 1 X Wash Buffer (PBST). After the last wash, the remaining wash buffer was removed carefully and completely by aspiration and blotting it against clean paper towels. The prepped, empty wells were blocked at room temperature for 1 hr by adding 300 µL of Reagent Diluent (20% BSA in 1 X PBST) to each well. Wash the wells as in the step above. Next we added the samples and standard protein of known increasing concentrations (100 µl in volume) diluted in Reagent Diluent. The plates were then covered with an adhesive strip and incubated for 2 hr at room temperature. Post incubation, the wells were washed as above for plate preparation. Following the washes, we added 100 µl of freshly prepared Detection Antibody diluted 100 times in PBS. The plate was covered and incubated for 2 hr at room temperature. Post incubation the wells were washed thrice as in the previous step. Finally for generating detectable signal, we added 100 µl of Streptavidin-HRP to each well, covered the plate and incubated it in dark at room temperature for 20 min. Next the wells were washed thrice as in steps above. After aspirating the wells clean of any buffer we added 100 µl of Substrate Solution to each well and incubated the plates again in dark at room temperature for 20 min. Finally, 50 µl of stop solution was added to each well and mixed well by gentle tapping. The signal was measured using a microplate reader set to 450 nm, 570, and 540 nm. Perform wavelength correction by subtracting readings at 540 nm or 570 nm from the readings at 450 nm.

## Antibodies
Primary antibodies

TRF2 rabbit polyclonal (Novus NB110-57130)- ChIP, IP, WB.
TRF2 mouse monoclonal (Millipore 4A794)- Flow cytometry.
TRF1 mouse monoclonal (Novus NB110-68281)- WB.
IL1R1 rabbit polyclonal (abcam ab106278)- IF, Flow-Cytometry, WB.
DDK/FLAG mouse monoclonal (Sigma-F1804)- ChIP, WB
BG4 recombinant antibody (Sigma Aldrich MABE917)- ChIP.
P300 (CST D2X6 N) rabbit monoclonal- ChIP, IF, WB.
Ac-p300/CBP (CST 4771) rabbit monoclonal- ChIP, WB.
CBP (CST 7389) rabbit monoclonal- ChIP, WB.
P65(NFKB) (CST 8242) rabbit monoclonal- WB.
Ser536 P p65 NFKB (CST 3033) rabbit monoclonal- WB
IKB (CST 4814) mouse monoclonal- WB.
p-IKB (CST 2589) rabbit monoclonal- WB.
CD44 (CST 3570) mouse monoclonal- IF.

Histone H3 rabbit polyclonal (Abcam ab1791)- ChIP.
H3K27ac rabbit polyclonal (Abcam ab4729)- ChIP.
H3K4me3 rabbit polyclonal (Abcam ab8580)- ChIP.
H3K27me3 rabbit monoclonal (abcam ab192985)- ChIP.
H3K9me3 rabbit polyclonal (abcam ab8898)- ChIP.
Beta-actin mouse monoclonal (Santacruz C4)- WB.
GAPDH mouse monoclonal (Santacruz 6C5)- WB.
CD11b Monoclonal Antibody (ICRF44), eBioscience- Flow cytometry.
CD206 Monoclonal Antibody (MR5D3) Invitrogen- Flow cytometry.
APC anti-human CD326 (EpCAM) (9C4) BioLegend- Flow cytometry.
FITC anti-human CD11B (ICRF44) BioLegend- Flow cytometry.
PE anti-human CD45 (HI30) BioLegend- Flow cytometry.
PE-Cy7 anti-human CD45 (W17233E) BioLegend- Flow cytometry.
Anti-Rabbit IgG/Anti-mouse IgG (Millipore) was used for isotype control.

### Secondary antibodies
Anti-Rabbit-HRP(CST), anti-Mouse-HRP(CST), mouse-ALP (Sigma), rabbit-ALP (Sigma), anti-rabbit Alexa Fluor 488, anti-mouse Alexa Fluor 594 (Molecular Probes, Life Technologies).

### Real-time PCR
Total RNA was isolated from cells using TRIzol Reagent (Invitrogen, Life Technologies) according to manufacturer's instructions. For TNBC tissue and xenograft, Allprep DNA/RNA Universal kit from QIAGEN was used. The cDNA was prepared using High-Capacity cDNA Reverse Transcription Kit (Applied Biosystems) using manufacturer provided protocol. A relative transcript expression level for genes was measured by quantitative real-time PCR using a SYBR Green (Takara) based method. Gene expression was calculated as $2^{-delCT}$ with delCT being the difference in threshold cycles (Ct) between test and internal control. *GAPDH* / 18 S gene was used as internal control for normalizing the cDNA concentration of each sample. For fold change of gene expression calculations, $2^{-delCT}$ for the first replicate of the control sample was used for normalisation for the samples. Normalised expression values (fold change) were plotted and compared with appropriate statistical testing.

RT PCR primers used have been enlisted as follows:

| mRNA expression primers | |
| --- | --- |
| IL1R1FP | GGCTGAAAAGCATAGAGGGAAC |
| IL1R1 RP | CTGGGCTCACAATCACAGG |
| TRF2 FP | CAGTGTCTGTCGCGGATTGAA |
| TRF2 RP | CATTGATAGCTGATTCCAGTGGT |
| IL1B FP | ATGATGGCTTATTACAGTGGCAA |
| IL1B RP | GTCGGAGATTCGTAGCTGGA |
| IL1A FP | AGATGCCTGAGATACCCAAAACC |
| IL1A RP | CCAAGCACACCCAGTAGTCT |
| IL2 FP | CATTTGTGGTTGGGTCAGG |
| IL2 RP | AGTGAGGAACAAGCCAGAGC |
| IL6 FP | AGCACTCCTTGGCAAAACTG |
| IL6 RP | CGGAAGGAACCATCTCACTG |
| TNF FP | GCCAGAGGGCTGATTAGAGA |
| TNF RP | TCAGCCTCTTCTCCTTCCTG |
| IL8 FP | GAATGGGTTTGCTAGAATGTGATA |

*Continued on next page*

*Continued*

| mRNA expression primers | |
|---|---|
| IL8 RP | CAGACTAGGGGTTGCCAGATTTAAC |
| IL10 FP | GACTTTAAGGGGTTACCTGGGTTG |
| IL10 RP | TCACATGCGCCTTGATGTCTG |
| IL4 FP | CCAACTGCTTCCCCCTCTG |
| IL4 RP | TCTGTTACGGTCAACTCGGTG |
| IL13 FP | GAAGGCTCCGCTCTGCAAT |
| IL13 RP | TCCAGGGCTGCACAGTACA |
| GAPDH FP | TGCACCACCAACTGCTTAGC |
| GAPDH RP | GGCATGGACTGTGGTCATGAG |
| 18 S FP | TTCGGAACTGAGGCCATGAT |
| 18 S RP | TTTCGCTCTGGTCCGTCTTG |

TRF2 target gene expression primers from *Mukherjee et al., 2018*.

## RNA-seq analysis

RNA was extracted using TRIzol TruSeq RNA Library Prep Kit v2 was used from preparation of libraries for sequencing. Raw Illumina sequencing reads were checked for quality using FastQC (version 0.11.9) followed by adapter clipping and trimming using Trimmomatic (version 0.39) with default parameters. Trimmed reads were then aligned to the human reference genome (GRCh38, GENCODE v36) using STAR aligner (version 2.7.8 a; *Dobin et al., 2013*). FeatureCounts (subread package version 2.0.1) was used to quantify gene expression (*Quinlan and Hall, 2010*). Quality checks were performed at each step using the MultiQC tool (version 1.10.1). Differential gene expression analysis was performed using the DESseq2 package (version 1.30.0; *Love et al., 2016*) in R (version 4.0.3). The analysis was performed by removing the effects of confounding variables such as batch and replicate using the appropriate design formula. Gene expression was normalised using variant stabilizing transformation (vst) for visualisation purposes. Genes with BH-adjusted p-value <0.05 and absolute Log2 fold change greater than 1 (at least 100% fold change in either direction) were taken as significantly differentially expressed. Two replicates for each sample were used for differential gene expression analysis.

## ChIP (chromatin immunoprecipitation)

ChIP assays were performed as as previously reported by *Mukherjee et al., 2019a*; *Mukherjee et al., 2019b*; *Mukherjee et al., 2018*. Briefly, 3–5 million cells were fixed with ~1% formaldehyde (Sigma-Aldrich) for 10 min and lysed. Chromatin was sheared to an average size of ~300–400 bp using Biorupter (Diagenode). 10% of the sonicated fraction was processed as input using phenol–chloroform and ethanol precipitation. ChIP was performed for endogenous/ectopically expressed protein using 1:100 dilution (v/v) of the respective ChIP grade antibody incubated overnight at 4 °C. Immune complexes were collected using Salmon sperm DNA-saturated Magnetic Dynabeads (–50 µg per sample) and washed extensively. Phenol-chloroform-isoamylalcohol was used to extract DNA from the immunoprecipitated fraction. ChIP DNA was quantified by Qubit HS ds DNA kit (Thermo Fisher Scientific). Quantified ChIP samples were validated by qRT-PCR.

ChIP enrichment was calculated over 1% input as: $2^{-(CT\ ChIP-CT\ 1\%\ input)}$. Following this, the ChIP enrichment was normalised to enrichment value for respective IgG (Mock) or total H3 ChIP.

The ChIP-qRT PCR primers are as follows:

| ChIP- qRT PCR primers | |
|---|---|
| IL1R1-1000 FP | CCCCTTCCACAGAAGAAACCTGG |
| IL1R1-1000 RP | CGCAGAAGCCAGTGGGAGG |

*Continued on next page*

*Continued*

| ChIP- qRT PCR primers | |
|---|---|
| IL1R1-800 FP | AGCCCAGGTGGGTGCCA |
| IL1R1-800 RP | ATGACAGAATGATTCTTTGCTCTGTGG |
| IL1R1-600 FP | TATTTGTCAGGCGCCCCTGG |
| IL1R1-600 RP | CATACATCTGTGTGTGCCTGGG |
| IL1R1-400 FP | CAGGAAGCCATACATCTGTGTAGC |
| IL1R1-400 RP | GAATGTTATACACACACTCATACACACACACA |
| IL1R1-200 FP | GAGTGTGTGTATAACATTCATTACTGCAAAC |
| IL1R1-200 RP | ATGAAAAGATACAAGTGAAGAGCTGCC |
| IL1R1+200 FP | TGGGAGGTGACACCCAGTTTAAG |
| IL1R1+200 RP | CCAACTCTGGGCCTTCCCAC |
| IL1R1 3' UTR FP | TCCTTTGACTTATTGTCCCCACT |
| IL1R1 3' UTR RP | TCTTCAACACTTAGGGCATCTT |
| IL1R1 –20 KB FP | TTGTCCTGAATAGCCGCAGG |
| IL1R1 –20 KB RP | CCCTCGAGAAAACAGCCCAT |

## Immunofluorescence microscopy

Adherent cells were seeded on coverslips and allowed to reach a confluency of ~70%. Cells were fixed using freshly prepared 4% Paraformaldehyde by incubating for 10 min at RT. Cells were permeabilised with 0.25% Triton X-100 (10 min at RT) and treated with blocking solution (5% BSA in PBS) for 2 hr at RT. All the above stated were followed by three washes with ice cold PBS for 5 min each. Post-blocking, cells treated with relevant antibodies (1:500 for TRF2, 1:500 for IL1R1 and 1:1000 for CD44/p300) and incubated overnight at 4 °C in a humid chamber. Post-incubation, cells were washed alternately with PBS and PBST three times and probed with secondary Ab (rabbit Alexa Fluor 488 (1:1000) / mouse Alexa Fluor 594 (1:1000)) for 2 hr at RT. Cells were washed again alternately with PBS and PBST three times and mounted with Prolong Gold anti-fade reagent with DAPI. Images were taken on Leica TCS-SP8 confocal microscope/ Cytiva Deltavision microscope. Lasers of emission wavelength 405 nm, 488 nm and 532 nm were used imaging for DAPI, Alexa Fluor 488 secondary Ab and Alexa Fluor 594 Ab respectively. Laser intensity was kept 20% for DAPI and 40–60% for Alexa Fluor secondary antibodies. ImageJ (FIJI) software was used to calculate signal intensity (a.u.).

## Western blotting

For western blot analysis, protein lysates were prepared by suspending cell pellets in 1 X RIPA (with 1 x mammalian protease inhibitor cocktail). Protein was separated using 10%/12% SDS-PAGE and transferred to polyvinylidenedifluoride membranes (Immobilon FL, Millipore). After blocking, the membrane was incubated with primary antibodies. Post incubation with primary antibodies (1:1000 v/v), the membrane was washed with 1 X PBS and then incubated with appropriate secondary antibodies (1:2000 v/v). Following secondary antibody incubation, the membrane was washed with 1XPBS. The blot was finally developed using HRP substrate/BCP NBT and reagents from Millipore.

## HT1080 TRF2 and TERT inducible cells

TRF2 cDNA sequence was inserted into pENTR11 vector and the cassette was transferred into lentiviral plasmid pCW57.1 by gateway cloning. Following this viral particles were generated from HEK293T cells and the viral soup with polybrene (Sigma-Aldrich) was used for transduction. Cells with

insert were selected with 1 µg/ml of puromycin and tested for TRF2 induction using varying doxycycline concentrations.

Similarly TERT cDNA was amplified from a mammalian expression vector and inserted into pENTR11 vector. Subsequently, the same strategy was used to generate and test TERT inducible HT1080 cells.

## Vector constructs

The pCMV6-myc-DDK(FLAG)-TRF2 vector (TRF2 WT) was procured from Origene (RC223601). The mutants delB and delM constructs used in the study have been previously reported (*Hussain et al., 2017*) and made through mutagenesis of the TRF2 WT construct. TRF2 shRNA was procured from Origene (TL308880). Side-directed Mutagenesis was performed on the TRF2 WT plasmid to generate the PTM mutants.

For customizing dCas9-TRF2 fusion plasmids, the backbone of dCAS9-VP64-GFP (Addgene 61422) was used. IL1R1 sgRNA was custom designed against IL1R1 promoter G-quadruplex (G4-motif B) and cloned in pX333 (Addgene 64073) backbone after replacing Cas9 with mCherry.

## Transfections

TRF2 wildtype (myc/DDK tag) or mutant mammalian expression vector pCMV6 was transfected into HT1080 /MDAMB231 cells that were 60% confluent using Lipofectamine 2000 transfection reagent (following the manufacturers' protocol). 2–4 µg of plasmid was used for transfection in a 35 mm well for each case.

In case of TRF2 shRNA (Origene)/TERC shRNA (Santa Cruz Biotechnology), 4 µg of plasmid was used for transfection in a 35 mm well for each case and Fugene HD transfection reagent was used as per manufacturer's protocol. Cells were then maintained in 1 µg/ml of puromycin for stable shRNA maintenance. For dCAS9-TRF2 fusion constructs, co-transfected with IL1R1 sgRNA plasmid, FugeneHD was used at recommended concentrations.

## TRF2 silencing by siRNA

HT1080 cells were transfected with TRF2 siRNA oligonucleotides (synthesised from Eurogenetics Pvt. Limited; *Sharma et al., 2021*) using lipofectamine 2000 (Invitrogen) transfection reagent according to manufacturer's instructions. Silencing was checked after 48 hr of transfection. Pooled scrambled siRNA was used as control.

## Luciferase assay

IL1R1 promoter (upto 1500 bp upstream of transcription start site) was cloned in PGL3 basic vector upstream of firefly luciferase construct. G4 motif disrupting mutations were introduced in the promoter construct by site directed mutagenesis.

Plasmid (pGL4.73) containing a SV-40 promoter driving Renilla luciferase was co-transfected as transfection control for normalisation. After 48 hr, cells were harvested and luciferase activities of cell lysate were recorded by using a dual-luciferase reporter assay system (Promega).

## Circular dichroism

Circular Dichroism (CD) profiles (220 nm to 300 nm) were obtained for representative G4 motifs identified within IL1R1 promoter and an overlap with TRF2 high confidence ChIP seqpeak. CD showed the formation of G4-motif with the unaltered sequence while mutated G4 sequence gave partial/complete disruption of the G4-motif under similar conditions (buffer used for G- quadruplex formation –10 mM Na cacodylate and 100 mM KCl). The CD spectra were recorded on a Jasco-89spectropolarimeter equipped with a Peltier temperature controller. Experiments were carried out using a 1 mm pathlength cuvette over a wavelength range of 200–320 nm. Oligonucleotides were synthesised commercially from Sigma-Aldrich. 2.5 µM oligonucleotides were diluted in sodium cacodylate buffer (10 mM sodium cacodylate and 100 mM KCl, pH 7.4) and denatured by heating to 95 °C for 5 min and slowly cooled to 15 °C for several hours. The CD spectra reported here are representations of three averaged scans taken at 20 °C and are baseline corrected for signal contributions due to the buffer.

## CCR5-IL1R1 promoter insert cells

IL1R1 promoter and downstream firefly luciferase (and G4 mutant variant) was inserted in HEK293T cells by using strategy previously standardised and reported for TERT promoter-gaussia insertion in the same locus (*Sharma et al., 2021*).

## IL1R1 promoter G4 mutant HEK cells

Custom designed gRNA pair was used to insert mutant G4 template sequence (G4 mutantB) into the IL1R1 endogenous promoter by co-transfecting Cas9and gRNA pair with doner template SSODN sequence. Following this, single cell screen was done for positive mutant colony. Once identified an appropriate mutant colony was used for experiments.

   sgRNA and SSODN details are as follows:
   sgRNA1- 5'- GCTGCCAATGGGT**G**GAGTCTTG**G**G –3' (efficiency score (Doench et al)- 35)
   -ve strand, ssODN: 91 bp (PAM proximal)- 36 bp (PAM distal)
   Target strand is +ve strand. Non-target stand is –ve strand. ssODN should be +ve strand.
   **ssODN** (mutated): chr2:102070276–102070402
   GGGACACTGCAGCCCTGGCCTGGCCTACTTTTCTCTTCTCCCATATCTGGAAAATGAAAGTAGCCTGACGTATCCGGGGACACGGCACAAGACTCAACCCATTGGCAGCTCTTCACTTGTATCTTTT
   sgRNA2- 5'- GCAATGGGT**G**GAGTCTTG**G**GCCG**G** –3' (efficiency score (Doench et al)- 40)
   -ve strand, ssODN: 85 bp (PAM proximal)- 42 bp (PAM distal).

## Immunoprecipitation

Immunoprecipitation of proteins was performed using modified version of suggested IP protocol from CST. Cells were lysed using cell lysis buffer ((1 X) 20 mM Tris (pH 7.5), 150 mM NaCl (Sigma-Aldrich), 1 mM EDTA (Sigma-Aldrich), 1 mM EGTA (Sigma-Aldrich), 1% Triton X-100 (Sigma-Aldrich), 2 X mPIC (Thermo Fisher Scientific)) and treated with relevant primary antibody (1: 100 v/v) overnight at 4 °C or mock IgG on rotor. 50 μg of Dyanbeads (blocked with BSA-5% for 60 mins at RT) was added to the lysis soup and kept for 2 hr in mild agitation. Dynabeads were magnetically separated and washed thrice (with 1 X PBS with 0.1% Tween-20). The beads were resuspended with 1 X RIPA (50 μl) and 1 X Laemmli buffer. This was followed by denaturation at 95°Cfor 10 min and followed by western blot for specific proteins.

## HAT assay

Histone acetyl transferase (HAT) assay was performed as per manufacturers' protocol from Active motif. Mammalian TRF2 and p300 full length were commercially procured from Origene and Abcam respectively and used for the assay.

## IL1R1 knockout HT1080 cells

Two sets of gRNAs targeting either exon 3 or exon3-7 of *IL1R1* coding region were cloned in PX459 (Addgene 62988) and co-transfected in HT1080 cells, followed by puromycin selection and single cell screening for IL1R1 knockout cells. A knockout clone was selected (based on IL1R1 depletion) and used for further experiments.

## Gene ontology and cBIO PORTAL analysis

Gene Ontology for KEGG pathway enrichment and associated figures were generated using ShinyGO web portal (*Ge et al., 2020*). Gene ontology (Biological Process) analysis was done using gene list (*Barthel et al., 2017*) using publicly available resource –The Gene Ontology Resource database (http://geneontology.org/). Fold enrichment, FDR values and p-values were calculated using PANTHER. Ontology plot was made using MS Excel multivariate plot. *IL1R1* high/low samples were segregated, and differential gene expression gene list was generated using cBioPortal (https://www.cbioportal.org/) using the TCGA Breast Invasive carcinoma dataset (Firehose legacy). Sample IDs for the patient samples have been provided in *Supplementary file 4*. For *IL1R1* high samples, the cut-off used was 1.5 times the standard deviation of the mean expression value. Custom query commands were used to fetch the *IL1R1* high and low sample IDs. DEG list and survival plots were generated using in-built functions of cBioPortal.

## TRAP (telomere repeat amplification protocol) for telomerase activity

Protein lysate was prepared in 1 X CHAPS Lysis Buffer and the assay was performed using TRAPeze RT Telomerase Detection Kit (Merck) per manufacturer's instructions.

## Statistical analysis

All statistical tests were performed using interface provided within GRAPHPAD PRISM software. Median values and correlations were calculated using MS excel functions.

## Acknowledgements

We acknowledge Manish Rai (confocal imaging), Debojyoti Chakraborty Lab (DeltaVision Microscopy), Vivek Rao (THP1 cells) and Mohit Agarwal (ELISA experiments) from CSIR-IGIB. Fellowships from CSIR (AKM, AS, SSR, SS, ASG, SVM, SB and DS); DBT-Wellcome Trust India Alliance (AKM, AS, MC, MV and AP); UGC (SD) and DST (DK) are acknowledged. RS and NG acknowledges intramural funding support from NCBS-TIFR. This work was supported by the DBT/Wellcome Trust India Alliance Fellowship [grant number IA/S/18/2/504021] awarded to SC, and research funding from the Council of Scientific and Industrial Research (CSIR), Ashoka University and Department of Biotechnology (DBT).

## Additional information

### Funding

| Funder | Grant reference number | Author |
|---|---|---|
| Wellcome Trust DBt India Alliance | IA/S/18/2/504021 | Ananda Kishore Mukherjee<br>Ankita Singh<br>Megha Chatterjee<br>Meenakshi Verma<br>Ahmad Perwez<br>Shantanu Chowdhury |
| National Centre for Biological Sciences | | Nija George<br>Radhakrishnan Sabarinathan |
| Tata Institute of Fundamental Research | | Nija George<br>Radhakrishnan Sabarinathan |
| Council for Scientific and Industrial Research | | Ananda Kishore Mukherjee<br>Ankita Singh<br>Shalu Sharma<br>Shuvra Shekhar Roy<br>Antara Sengupta<br>Soujanya Vinayagamurthy<br>Sulochana Bagri<br>Dristhi Soni |
| University Grants Commission | | Subhajit Dutta |
| Department of Biotechnology, Ministry of Science and Technology, India | | Shantanu Chowdhury |
| Ashoka University | | Shantanu Chowdhury |

The funders had no role in study design, data collection and interpretation, or the decision to submit the work for publication. For the purpose of Open Access, the authors have applied a CC BY public copyright license to any Author Accepted Manuscript version arising from this submission.

## Author contributions
Ananda Kishore Mukherjee, Conceptualization, Resources, Data curation, Formal analysis, Validation, Investigation, Visualization, Methodology, Writing – original draft, Project administration, Writing – review and editing; Subhajit Dutta, Data curation, Formal analysis, Validation, Investigation, Visualization, Methodology; Ankita Singh, Shuvra Shekhar Roy, Antara Sengupta, Megha Chatterjee, Sulochana Bagri, Resources, Data curation, Investigation, Methodology; Shalu Sharma, Resources, Investigation, Methodology; Soujanya Vinayagamurthy, Data curation, Investigation, Methodology; Divya Khanna, Ahmad Perwez, Resources, Methodology; Meenakshi Verma, Investigation, Methodology; Dristhi Soni, Resources; Anshul Budharaja, Formal analysis, Visualization, Methodology; Sagar Kailasrao Bhisade, Formal analysis, Methodology; Vivek Anand, Nija George, Mohammed Faruq, Ishaan Gupta, Radhakrishnan Sabarinathan, Formal analysis, Investigation, Methodology; Shantanu Chowdhury, Supervision, Funding acquisition, Project administration, Writing – review and editing

## Author ORCIDs
Ananda Kishore Mukherjee ⬥ https://orcid.org/0000-0002-3082-6749
Soujanya Vinayagamurthy ⬥ https://orcid.org/0000-0003-1465-9925
Sulochana Bagri ⬥ https://orcid.org/0000-0001-7407-2558
Shantanu Chowdhury ⬥ https://orcid.org/0000-0001-7185-8408

## Ethics
Human TNBC tissue samples were procured after due institutional ethics committee clearance as follows: 1. Rajiv Gandhi Cancer Centre and Research Institute-Delhi, India - RGCIRC/IRB/119/2020 clearance for project awarded to Dr. Ankita Singh and Dr. Shantanu Chowdhury under the project titled- "molecular mechanisms of how telomeres impact global gene expression : potential role of TRF2 in telomere-chromatin interactions", 2. CSIR-Institute of Genomics and Integrative Biology-Delhi, India - Human ethics committee clearance awarded to Dr. Shantanu Chowdhury (CSIR-IGIB/IHEC/20219-20/ dt. 04.02.2020) for the project titled- "molecular mechanisms of how telomeres impact global gene expression : potential role of TRF2 in telomere-chromatin interactions". All committee guidelines were followed for the procurement and experiments with fixed TNBC tissue samples. All informed content from patients was obtained following committee recommendations.

The animal experiments were performed in accordance with guidelines and recommendations laid out by the institutional animal ethics committee (IAEC) at CSIR-Institute of Genomics and Integrative Biology-Delhi, India after due clearance was awarded to Dr. Shantanu Chowdhury (IGIB/ IAEC/11/28/2020).

Reviewer #2 (Public review): https://doi.org/10.7554/eLife.95106.3.sa1
Reviewer #3 (Public review): https://doi.org/10.7554/eLife.95106.3.sa2
Author response https://doi.org/10.7554/eLife.95106.3.sa3

# Additional files

## Supplementary files
MDAR checklist

Supplementary file 1. DEG list (upregulated and downregulated) for long vs short telomere HT1080 fibrosarcoma cells.

Supplementary file 2. Immune related TRF2 ChIP-seq target genes.

Supplementary file 3. RTL for TNBC samples (normalized to HT1080).

Supplementary file 4. TCGA sample list for IL1R1 high vs low BRCA analysis.

Supplementary file 5. Relative expression and correlation of key immune marker genes in BRCA patient sample data curated from TCGA.

## Data availability
The authors declare that all data necessary to support the inferences are included in the manuscript and supplementary files. Source data has been provided for all figures and figure supplements.

Additionally, raw data for the RNA sequencing (HT1080-HT1080-LT cells) has been made public (GEO accession code GSE267781).

The following dataset was generated:

| Author(s) | Year | Dataset title | Dataset URL | Database and Identifier |
|---|---|---|---|---|
| Chowdhury S, Mukherjee AK, Sharma S, Faruq M, Budharaja A, Gupta I | 2024 | Transcriptional outcome of telomere elongation in human fibrosarcoma model | https://www.ncbi.nlm.nih.gov/geo/query/acc.cgi?acc=GSE267781 | NCBI Gene Expression Omnibus, GSE267781 |

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
