## [Editor Report · eLife Assessment]

This study presents an **important** finding on the role of telomeres in modulating interleukin-1 signaling and tumor immunity in TNBC. The evidence supporting these findings is **solid**, presented through comprehensive analyses including TNBC clinical samples, tumor-derived organoids, cancer cells, and xenografts. The work will be of broad interest to cell and medical biologists focusing on TNBC.

---

## [Referee Report · Reviewer #2 (Public review)]

This study highlights the role of role of telomeres in modulating IL-1 signaling and tumor immunity. The authors demonstrate a strong correlation between telomere length and IL-1 signaling by analyzing TNBC patient samples and tumor-derived organoids. Mechanistic insights revealed that non-telomeric TRF2 binding at the IL-1R1. The observed effects on NF-kB signaling and subsequent alterations in cytokine expression contribute significantly to our understanding of the complex interplay between telomeres and the tumor microenvironment. Furthermore, the study reports that the length of telomeres and IL-1R1 expression is associated with TAM enrichment. However, the manuscript lacks in-depth mechanistic insights into how telomere length affects IL-1R1 expression Overall, this work broadens our understanding of telomere biology.

---

## [Referee Report · Reviewer #3 (Public review)]

Summary:

In this manuscript, entitled "Telomere length sensitive regulation of Interleukin Receptor 1 type 1 (IL1R1) by the shelterin protein TRF2 modulates immune signalling in the tumour microenvironment", Dr Mukherjee and colleagues pointed at clarifying the extra-telomeric role of TRF2 in regulating IL1R1 expression with consequent impact on TAMs tumor-infiltration.

Strengths:

Upon a careful manuscript evaluation, I feel to conclude that the presented story is undoubtedly well conceived. At technical level, experiments have been properly performed and the obtained results well-support author conclusions.

Weaknesses:

Unfortunately, the covered topic is not particularly novel. In detail, TRF2 capability of binding extratelomeric foci in cells with short telomeres has been well demonstrated in a previous work published by the same research group. The capability of TRF2 to regulate gene expression is well-known, the capability of TRF2 to interact with p300 has been already demonstrated and, finally, the capability of TRF2 to regulate TAMs infiltration (that is the effective novelty of the manuscript) appears as an obvious consequence of IL1R1 modulation (this is probably due to the current manuscript organization).

---

## [Author Response]

The following is the authors’ response to the original reviews.

**Reviewer #1 (Public Review):**
Summary:This manuscript from Mukherjee et al examines potential connections between telomere length and tumor immune responses. This examination is based on the premise that telomeres and tumor immunity have each been shown to play separate, but important, roles in cancer progression and prognosis as well as prior correlative findings between telomere length and immunity. In keeping with a potential connection between telomere length and tumor immunity, the authors find that long telomere length is associated with reduced expression of the cytokine receptor IL1R1. Long telomere length is also associated with reduced TRF2 occupancy at the putative IL1R1 promoter. These observations lead the authors towards a model in which reduced telomere occupancy of TRF2 - due to telomere shortening - promotes IL1R1 transcription via recruitment of the p300 histone acetyltransferase. This model is based on earlier studies from this group (i.e. Mukherjee et al., 2019) which first proposed that telomere length can influence gene expression by enabling TRF2 binding and gene transactivation at telomere-distal sites. Further mechanistic work suggests that G-quadruplexes are important for TRF2 binding to IL1R1 promoter and that TRF2 acetylation is necessary for p300 recruitment. Complementary studies in human triple-negative breast cancer cells add potential clinical relevance but do not possess a direct connection to the proposed model. Overall, the article presents several interesting observations, but disconnection across central elements of the model and the marginal degree of the data leave open significant uncertainty regarding the conclusions.Strengths:Many of the key results are examined across multiple cell models.The authors propose a highly innovative model to explain their results.Weaknesses:Although the authors attempt to replicate most key results across multiple models, the results are often marginal or appear to lack statistical significance. For example, the reduction in IL1R1 protein levels observed in HT1080 cells that possess long telomeres relative to HT1080 short telomere cells appears to be modest (Supplementary Figure 1I). Associated changes in IL1R1 mRNA levels are similarly modest.Related to the point above, a lack of strong functional studies leaves an open question as to whether observed changes in IL1R1 expression across telomere short/long cancer cells are biologically meaningful.Statistical significance is described sporadically throughout the paper. Most major trends hold, but the statistical significance of the results is often unclear. For example, Figure 1A uses a statistical test to show statistically significant increases in TRF2 occupancy at the IL1R1 promoter in short telomere HT1080 relative to long telomere HT1080. However, similar experiments (i.e. Figure 2B, Figure 4A - D) lack statistical tests.

TRF2 overexpression resulted in ~ 5-fold or more change in *IL1R1* expression. Compared to this, telomere length-dependent alterations in *IL1R1* expression, although about 2-fold, appear modest (~ 50% reduction in cells with long telomeres across different model systems used). Notably, this was consistent and significant across cell-based model systems and xenograft tumors (see Figure 1). Unlike TRF2 induction, telomere elongation or shortening vary within the permissible physiological limits of cells. This is likely to result in the observed variation in *IL1R1* levels.

For biological relevance, we have shown this using multiple models where telomere length was either different (patient tissue, organoids) or were altered (cell lines, xenograft models) . Where IL1 signalling in TNBC tissue and tumor organoids, and cells/xenografts were shown to impact M2 macrophage infiltration in a telomere length sensitive fashion. We made use of the tumor organoids to test M2 macrophage infiltration using IL1RA and small molecule based *IL1R1* inhibition.

We have now included statistical tests in all the relevant figures and incorporated the necessary details about the tests performed in the figure legend for clarity of readers. Additionally, all data points, p values and details of statistical tests have been included in Figure wise excel sheets for both main and supplementary figures.

**Reviewer #1 (Recommendations For The Authors):**
There are typos throughout the manuscript. The word 'expression' is incorrectly spelled on y-axis labels throughout the manuscript (for example see Figure 1B). The word 'telomere' is incorrectly spelled in Supplementary Figure 1 legend panel A. Most errors, such as these, do not interfere with my comprehension of the manuscript. However, others made the manuscript difficult to follow. For example, I think that MDAMB231, MDAMD231, and MDAM231 are frequently used interchangeably to refer to the same cell line. This makes it very difficult to understand certain experiments.I often found it difficult to understand which statistical test was used for a specific experiment. I suggest changing the style in the legends to more clearly connect statistical tests with specific data points.

We thank the reviewer for pointing out the typological errors. We have now made relevant corrections to both figures and text.

As stated above, we have now provided details of statistical tests performed in the figure legend for clarity of readers. Additionally, all data points, p values and details of statistical tests have been included in Figure wise excel sheets for both main and supplementary figures.

**Reviewer #2 (Public Review):**
This study highlights the role of telomeres in modulating IL-1 signaling and tumor immunity. The authors demonstrate a strong correlation between telomere length and IL-1 signaling by analyzing TNBC patient samples and tumor-derived organoids. Mechanistic insights revealed non-telomeric TRF2 binding at the IL-1R1. The observed effects on NF-kB signaling and subsequent alterations in cytokine expression contribute significantly to our understanding of the complex interplay between telomeres and the tumor microenvironment. Furthermore, the study reports that the length of telomeres and IL-1R1 expression is associated with TAM enrichment. However, the manuscript lacks in-depth mechanistic insights into how telomere length affects IL-1R1 expression. Overall, this work broadens our understanding of telomere biology.

The mechanism of how telomere length affects IL1R1 expression involves sequestration and reallocation of TRF2 between telomeres and gene promoters (in this case, the IL1R1 promoter). We have previously shown this across multiple genomic sites (Mukherjee et al, 2018; reviewed in J. Biol. Chem. 2020, Trends in Genetics 2023). We have described this in the manuscript along with references citing the previous works. A scheme explaining the model was provided as Additional Supplementary Figure 1, along with a description of the mechanistic model.

Figure 1-4 in main figures describe the molecular mechanism of telomere-dependent IL1R1 activation. This includes ChIP data for TRF2 on the IL1R1 promoter in long/short telomeres, as well as TRF2-mediated histone/p300 recruitment and IL1R1 gene expression. We further show how specific acetylation on TRF2 is crucial for TRF2-mediated IL1R1 regulation (Figure 5).

**Reviewer #2 (Recommendations For The Authors):**
The study primarily provides a snapshot of cytokine expression and telomere length at a single time point. Longitudinal studies or dynamic analyses could provide a more comprehensive understanding of the temporal relationship between telomere length and cytokine expression.Tumor heterogeneity is a significant problem for the various therapies. The study notes significant heterogeneity in telomere length but does not investigate the implications of this heterogeneity. Understanding the role of telomere length variation in different tumor cell populations is essential for a comprehensive interpretation of the results.The study only mentions a correlation between IL1R1 and relative telomere length but does not provide any potential clinical correlations with patient outcomes or survival. Addressing the clinical relevance of these molecular changes would improve the translational impact.

The importance of IL1R1 in prognostic and clinical outcomes of TNBC has been studied by multiple groups. The overall consensus is that higher IL1R1 leads to poor prognosis – aiding both cancer progression and metastasis. Using publicly available TCGA data, we found that IL1R1 high samples had significantly lower survival in breast cancer (BRCA) datasets. The results have now been included in the manuscript as Supplemnetray Figure 7G.

Addition in text:

“We, next, used publicly available TCGA gene expression data of breast cancer samples (BRCA) (**Supplementary file 4**) to assess the effect of *IL1R1* expression on cancer prognosis. We categorized samples based on *IL1R1* expression: *IL1R1* high (N=254) and *IL1R1* low samples (N = 709). It was seen that overall patient survival was significantly lower in *IL1R1* high samples (Log-rank p value -0.0149) (Supplementary Figure 7G). We also checked the frequency of occurrence of various breast cancer sub-types in *IL1R1* high and low samples (Supplementary Figure 7H). While invasive mixed mucinous carcinoma (the most abundant sub-type) was predominantly seen in *IL1R1* low samples, metaplastic breast cancer was only found within the *IL1R1* high samples. Interestingly, metaplastic breast cancer has been frequently found to be ‘triple negative’-i.e., ER-,PR- and HER2-. (Reddy et al., 2020).”

However, we could not access a TNBC (or any breast cancer dataset) that has been characterized for telomere length. Unfortunately, the clinical TNBC samples that we had access to did not have any paired short-term/long-term survival datasets. We could, in principle, use TERT/TERC expression as a proxy for telomere length; however, in our experiments, we found that telomerase activity did not positively correlate with telomere length as expected (Supplementary Figure 7C, Supplementary Figure 8D). Therefore, transcriptional signature (of telomere-associated genes) may not be a reliable indicator of telomere length.

The study lacks in-depth mechanistic insights into how telomere length affects IL1R1 expression and subsequently influences TAM infiltration. Further molecular studies or pathway analyses are necessary to elucidate the underlying mechanisms.

The mechanism involves sequestration and reallocation of TRF2 between telomeres and gene promoters (in this case, IL1R1 promoter). We have previously shown this across multiple genomic sites (Mukherjee et al, 2018). We have appropriately discussed this in the manuscript.

A schematic explaining the model has been provided as Additional Supplementary Figure 1.

We have provided ChIP data for TRF2 on IL1R1 promoter in long/short telomeres in the manuscript as well as histone/p300 ChIP and gene expression (Figure 1-4 in main figures exclusively deal with molecular mechanism of telomere dependent IL1R1 activation). We further go on to show how specific acetylation on TRF2 might be crucial for TRF2-mediated IL1R1 regulation (Figure 5). One of the key findings herein is the fact that TRF2 can directly regulate *IL1R1* expression through promoter occupancy- tested in telomere altered cell lines (HT1080, MDAMB231) and tumor xenografts (Figure 1 A, F, I- for TRF2 promoter occupancy).

Pathway analysis of HT1080 (short vs long telomere) transcriptome, shows that cytokine-cytokine receptor interaction is one of the key pathways in upregulated genes.

While we have focused on TRF2 mediated *IL1R1* regulation, it is quite possible that there are other telomere sensitive pathways/mechanisms by which *IL1R1* is regulated. This has been duly acknowledged in the discussion.

The manuscript title suggests modulation of immune signaling in the tumor microenvironment, yet the authors exclusively focus on CD206+ TAMs, limiting the scope. It is recommended to investigate other immune cell types for a more comprehensive understanding of changes in the immune tumor microenvironment.

As stated above, we approached the manuscript from the purview of TRF2-mediated IL1R1 regulation. In our assessment of TCGA data for breast cancer, we found that CD206 (MRC1) had the highest enrichment in *IL1R1* high samples among key TAM and TIL markers- now added as Figure 8A (Details in Supplementary file 5). It also had the highest correlation with *IL1R1* among the tested markers. Therefore, we proceeded to check CD206+ve TAMs.

Now the following section has been added to text:

“We further found that the total proportion of immune cells (% of CD45 +ve cells) did not vary significantly between short and long telomere TNBC samples (Supplementary Figure 8C). However, TNBC-ST samples had a higher percentage of myeloid cells (CD11B +ve) within the CD 45 +ve immune cell population. We checked in three TNBC-ST and TNBC-LT samples each and found that the percentage of M1 macrophages (CD86 high CD 206 low) in the myeloid population was lower than that of the M2 macrophages (CD 206 high CD 86 low) and unlike the latter, did not vary significantly between the TNBC-ST and TNBC-LT samples (Supplementary Figure 8C).”

Unfortunately, due to sample limitations we are unable to test this on a larger cohort of samples.

A single cell transcriptome experiment may have been a good way to have a more comprehensive immune profiling. However, with our TNBC samples, isolated nuclei for downstream processing had low viability as per 10X genomics specifications.

Does IL1R1 influence TAM recruitment or polarization within the tumor microenvironment? To assess the impact, the authors should use a marker indicative of M1-like macrophages, such as CD80 or CD86.

To address the issue of TAM recruitment vs polarization meaningfully we need to characterize tissue resident macrophages as well as macrophages in circulation. We did not have access to patient blood. A murine breast cancer in-vivo model might be a more appropriate model to test this, which would take considerable time for us to develop. It is something that we hope to address in a follow up study.

Did the authors analyze other breast cancer subtypes for telomere length?

Unfortunately, other breast cancer sub-types besides TNBC were not available to us for experimentation.

Figure legends are very briefly written and need to be elaborated. Scale bars are also missing in images.Add a gating strategy for flow cytometry results in Figure 8A.

Figure legend have been expanded for clarity. More prominent scale bars have been added for better visibility and reference. A relevant gating strategy has been added as Supplementary figure 8B.

**Reviewer #3 (Public Review):**
Summary:In this manuscript, entitled "Telomere length sensitive regulation of Interleukin Receptor 1 type 1 (IL1R1) by the shelterin protein TRF2 modulates immune signalling in the tumour microenvironment", Dr. Mukherjee and colleagues pointed out clarifying the extra-telomeric role of TRF2 in regulating IL1R1 expression with consequent impact on TAMs tumor-infiltration.Strengths:Upon careful manuscript evaluation, I feel that the presented story is undoubtedly well conceived. At the technical level, experiments have been properly performed and the obtained results support the authors' conclusions.Weaknesses:Unfortunately, the covered topic is not particularly novel. In detail, the TRF2 capability of binding extratelomeric foci in cells with short telomeres has been well demonstrated in a previous work published by the same research group. The capability of TRF2 to regulate gene expression is well-known, the capability of TRF2 to interact with p300 has been already demonstrated and, finally, the capability of TRF2 to regulate TAMs infiltration (that is the effective novelty of the manuscript) appears as an obvious consequence of IL1R1 modulation (this is probably due to the current manuscript organization).

Here we studied the TRF2-IL1R1 regulatory axis (not reported earlier by us or others) as a case of the telomere sequestration model that we described earlier (Mukherjee et al., 2018; reviewed in J. Biol. Chem. 2020, Trends in Genetics 2023). This manuscript demonstrates the effect of the TRF2-IL1R1 regulation on telomere-sensitive tumor macrophage recruitment. To the best of our knowledge, no previous study connects telomeres of tumor cells mechanistically to the tumor immune microenvironment. Here we focused on the IL1R1 promoter and provided mechanistic evidence for acetylated-TRF2 engaging the HAT p300 for epigenetically altering the promoter. This mechanism of TRF2 mediated activation has not been previously reported. Further, the function of a specific post translational modification (acetylation of the lysine residue 293K) of TRF2 in IL1R1 regulation is described for the first time. Additional experiments showed that TRF2-acetylation mutants, when targeted to the IL1R1 promoter, significantly alter the transcriptional state of the IL1R1 promoter. To our knowledge, the function of any TRF2 residue in transcriptional activation had not been previously described. Taken together, these demonstrate novel insights into the mechanism of TRF2-mediated gene regulation, that is telomere-sensitive, and affects the tumor-immune microenvironment.

We considered the reviewer’s suggestion to reorganize the result section. Reorganizing the manuscript to describe the TAM-related results first would, in our opinion, limit focus of the new findings and discovery [and novelty of the mechanisms (as described in above response, and in response to other comments by reviewers)] of the non-telomeric TRF2-mediated *IL1R1* regulation. We have tried to bring out the novelty, implications and importance of the TAM-related observations in the discussion.

**Reviewer #3 (Recommendations For The Authors):**
Based on the comments reported above, I would encourage the author to modify the manuscript by reorganizing the text. I would suggest starting from the capability of TRF2 to modulate macrophages infiltration. Data relative to IL1R1 expression may be used to explain the mechanism through which TRF2 exerts its immune-modulatory role. This, in my view, would dramatically strengthen the presented story.Concerning the text, "results" should be dramatically streamlined and background information should be just limited to the "introduction" section.

The manuscript should be carefully revisited at grammar level. A number of incomplete sentences and some typos are present within the text.

We thank the reviewer for the appreciation of our work for its technical strengths.

At the onset, we agree that we have explored the TRF2-IL1R1 regulatory axis. This underscores the significance of the telomere sequestration model that we had proposed earlier (Mukherjee et al., 2018). Herein, however, we significantly extend our previous work (which was more general and intended for putting forward the idea of telomere-dependent distal gene expression) by studying TRF2-mediated regulation of IL1 signalling (which was previously unreported). In addition, mechanistic details of how telomeres are connected to IL1 signaling through non-telomeric TRF2 are entirely new, not reported before by us or others.

We have removed some text descriptions from the result section to streamline the section.